



# Satellite observations of seasonality and long-term trend in cirrus cloud properties over Europe: Investigation of possible aviation impacts

Qiang Li  and Silke Groß

 Deutsches Zentrum für Luft- und Raumfahrt, Institut für Physik der Atmosphäre, D-82234 Oberpfaffenhofen, Germany,

**Correspondence:** Qiang Li (qiang.li@dlr.de)

**Abstract.** Linear contrails and contrail cirrus induced by global aviation have long been known to contribute to climate change by warming the atmosphere. Besides increasing global cirrus cloudiness, aviation may change the properties of the natural cirrus clouds by soot emissions which leads to increased heterogeneous freezing. In the first COVID-19 lockdown in Europe, changes in cirrus cloud properties and occurrence were detected with the lidar measurements of CALIPSO, which is supposed
to be caused by the reduction in civil aviation accordingly. In the last 10 years before COVID, however, aviation grew strongly in terms of CO2 emissions and flight densities in Europe. In this study, 10-year lidar measurements of cirrus clouds with CALIPSO are analyzed to determine the seasonality and long-term trends in cirrus clouds as well as their correlations with the ambient temperatures and air traffic. Cirrus clouds follow a distinct seasonal cycle in their occurrence rate (OR) and particle linear depolarization ratio (PLDR) $\delta_p$. Cirrus clouds appear within a broader altitude range in winter than in summer
and they are characterized by larger OR and $\delta_p$ values in winter than in summer. The monthly medians of $\delta_p$ as well as the deseasonalized time series of them in the 10-year period before COVID show both positive trends which are statistically significant according to the Mann-Kendall (MK) significance test. However, the cirrus occurrence shows a negative trend, which might be connected with the background meteorological conditions. Since the cirrus $\delta_p$ strongly depends on the ambient temperatures in cirrus, we further remove the contributions induced by temperatures from the cirrus $\delta_p$ with a simple linear
regression model. The derived residuals show significant positive trends with the MK test. To compare the cirrus $\delta_p$ and the air traffic densities, the deseasonalization of the data have previously been conducted since the seasonal cycles in both are not consistent. The deseasonalized time series of the cirrus $\delta_p$ and CO2 emissions from aviation both show an increasing trend and their correlation coefficients are $r = 0.54$ at the confidence level above 99.5%. Finally, the comparisons between the cirrus $\delta_p$ and aviation in every season were carried out and revealed a strong correlation in other seasons than in summer.

## 1  Introduction

Aviation affects the Earth's radiation budget through a combination of aviation emissions of CO2 and non-CO2 effects which have a warming effect on the atmosphere (Lee et al., 2021). Linear contrails and contrail cirrus induced by water vapor and soot emissions from air traffic in the upper atmosphere are expected to contribute a large part of the climate impact of aviation (e.g., Burkhardt and Kärcher, 2011). Due to their climate impact and a suitable tool to monitor the mitigation of aviation,





numerous experimental and theoretical efforts have been carried out in the last years to understand contrails and to determine their climate effect by calculating radiative forcing (e.g., Heymsfield et al., 2010; Voigt et al., 2011; Burkhardt and Kärcher, 2011; Graf et al., 2012; Schumann and Graf, 2013; Kärcher et al., 2015; Kärcher, 2018; Bock and Burkhardt, 2019; Schumann et al., 2021a, b). However, aviation emissions and the resulting contrail formation interact with the atmosphere through a complex manner which is still not fully understood.

Cirrus clouds, composed of ice crystals, usually appear in the upper atmosphere above ∼6 km with a variety of forms and shapes. They have long been recognized, as studies reveal that cirrus clouds permanently cover on average 30% of the Earth's surface with up to 70% coverage over the tropics and have a large impact on the Earth's radiation balance and climate evolution (Liou., 1986; Wang et al., 1996; Wylie and Menzel, 1999; Sassen and Campbell, 2001; Sassen et al., 2008; Nazaryan et al., 2008). Cirrus clouds influence the radiation balance by trapping the outgoing long-wave radiation from the ground

and underlying atmosphere (warming) and reflecting the incoming short-wave solar radiation back into space (cooling). The contribution of these two opposite effects depends on the cloud microphysical, thermal, and optical properties, that include cloud heights, temperatures, and the shape and orientation of ice crystals (e.g., Fu and Liou, 1993; Zhang et al., 1999; Zerefos et al., 2003; Stephens et al., 2004; Campbell et al., 2016). Overall, cirrus clouds are assumed to have a net warming effect of the climate system other than the low and midlevel clouds (Chen et al., 2000).

Midlatitude cirrus clouds are of particular interest and have been intensively studied thanks to the high number of observation sites as well as campaign-based observations (e.g., Voigt et al., 2017, 2022) at these latitudes. Nevertheless, midlatitude cirrus clouds are induced and affected by various weather patterns and interact with the atmospheric dynamics which lead to the difficulties of their representation in global and regional climate models (Boucher et al., 2013). Furthermore, midlatitude cirrus clouds can be strongly influenced by the frequent air traffic. Besides increasing global cirrus cloudiness (Boucher, 1999;

Minnis et al., 2004; Stubenrauch and Schumann, 2005), aviation-induced contrails and contrail cirrus also alter the optical and microphysical properties of natural cirrus clouds (e.g., Tesche et al., 2016; Urbanek et al., 2018; Li and Groß, 2021). Studies reveal that the effect of contrails and cirrus clouds on our climate is growing significantly, which is, however, still challenging the atmosphere community to parameterize their overall effect well (Bock and Burkhardt, 2019; Lee et al., 2021).

The radiative effects of cirrus clouds strongly depend on their microphysical properties, e.g., particle number concentration,

size, and shape (e.g., Stephens et al., 1990; Haag and Kärcher, 2004) which are further influenced by the ambient conditions (e.g., temperature and supersaturation) and the nucleation mode (Heymsfield, 1977; Khvorostyanov and Sassen, 1998; Ström and Ohlsson, 1998; Urbanek et al., 2018). Hence, an accurate estimate of ice crystal shape and orientation is very important for the radiative transfer simulation. In the last decades, both laboratory experiments and field observations reveals a high variety of habit diagrams of atmospheric ice crystals which are closely correlated with temperature and ice supersaturation (Bailey and

Hallett, 2004, 2009; Lawson et al., 2006, 2019). In natural clouds, however, ice crystals encounter varying temperature and humidity and may grow into irregular forms (Korolev et al., 1999, 2000). Furthermore, the depdendence of ice crystal habits on temperature is also governed by mass transport (including convection and advection) and the origin region of particles (e.g., Bailey and Hallett, 2009; Um et al., 2015).



It is well known that light scattered by atmospheric ice crystals may exhibit different polarization states from the incident
light. The computation of geometric ray tracing technique reveals that changes in polarization states depend on the internal
ray paths and, more precisely, increase with increasing hexagonal axis ratio (= length over width) (e.g., Takano and Liou,
1989). The particle linear depolarization ratio (PLDR) $\delta_p$ is a well-defined parameter to evaluate this effect and is widely used
to retrieve information on ice crystal habits, i.e., particle phase, shape, and orientation. In traditional lidar applications, $\delta_p$ is
defined as the ratio of power from both polarization components perpendicular and parallel to the transmitted laser source and
can be calculated using Equation (1)

$$\delta_p = \frac{\beta_\perp}{\beta_\parallel} \tag{1}$$

where $\beta_\perp$ and $\beta_\parallel$ are the perpendicular and parallel components of the backscatter coefficients retrieved from the ice crystals in
clouds, respectively. The light emitted by lidar exhibits the same orientation of polarization as the incident light if it is scattered
by spherical particles and different polarization if scattered by non-spherical particles such as cirrus ice crystals (Sassen et al.,
1989; Freudenthaler et al., 2009; Urbanek et al., 2018). As a well-established technique, polarization lidar has been widely
used to provide information on aerosol profiling and to distinguish between different types of aerosols, e.g., non-spherical
mineral dust particles with high values of $\delta_p$ (Freudenthaler et al., 2009; Tesche et al., 2009; Groß et al., 2012, 2013, 2015).
Further, this technique is also applied to unambiguously determine the cloud phase (e.g., Bühl et al., 2016) and to study ice
cloud properties (e.g., Schotland et al., 1971; Sassen, 1991; Ansmann et al., 2003; Groß et al., 2012; Urbanek et al., 2018).
$\delta_p$ is mainly determined by ice crystal shape that is a function of temperature, supersaturation (humidity), and potentially the
availability of ice nuclei during ice formation (e.g. Bailey and Hallett, 2004, 2009; Schnaiter et al., 2012; Järvinen et al., 2018).
$\delta_p$ is a suitable parameter used to retrieve information on ice crystal shape and hence to trace the aviation effects on the clouds.
Previous studies show that contrails and contrail cirrus are characterized with higher values of $\delta_p$ than persistent cirrus clouds
(Freudenthaler et al., 1996; Mishchenko and Sassen, 1998; Noel et al., 2006; Iwabuchi et al., 2012). In addition, Urbanek et al.
(2018) presented that cirrus clouds with enhanced $\delta_p$ values were found to be incidental to significantly lower supersaturations
inside the clouds indicating more frequent heterogeneous freezing. They also carried out a backward-trajectory analysis and
found that the affected ice clouds appear within areas under the influence of high aviation emissions. Under soot emissions ice
nucleation took place at lower supersaturation, which influences the form and size of ice crystals and further alters their optical
properties (Urbanek et al., 2018). During the first COVID-19 lockdown in Europe starting from mid-March of 2020, civil air
traffic was significantly reduced up to 88% in April 2020 compared to the previous year (e.g., Li and Groß, 2021). Based on
the analysis of lidar measurements with CALIPSO, a significant reduction in the cirrus $\delta_p$ was found in both March and April
2020 compared to the corresponding periods in the previous 6 years (Li and Groß, 2021). It is known that, however, the global
air traffic has been actually growing in the last decades in terms of the number of flights and CO2 emissions from aviation
before the COVID-19 pandemic, which, of course, is considered as a smaller change in air traffic compared to that caused by
the COVID-19 lockdown (see Figure 1 for the evolution of civil aviation in 42 European countries and regions in the last 11
years from 2010 to 2020). In this study, we extend the analyses of Li and Groß (2021) and study the possible aviation impact



on the microphysical properties of cirrus clouds in terms of seasonality and long-term trend in the occurrence rate and $\delta_p$ of cirrus clouds.

In Sect. 2 we will outline the CALIPSO data and methods. Sect. 3 describes our results concerning seasonal variations
and long-term trends in cirrus cloud properties and occurrence based on 10-year lidar measurements from March 2010 to February 2020. The dependence of the cirrus cloud properties on the corresponding ambient temperatures as well as aviation are determined and discussed in Sect.4. Our conclusions are finally summarized in Sect.5.

## 2 Data and methods

In the current study, we analyzed the measurements conducted with the CALIOP (Cloud-Aerosol Lidar with Orthogonal
Polarization) lidar which is carried aboard the CALIPSO satellite within the A-Train constellation in a sun-synchronous polar orbit (Winker et al., 2010; Stephens et al., 2018). CALIOP is a dual-wavelength elastic backscatter lidar at 532 and 1064 nm and polarization-sensitive at 532 nm (Winker et al., 2007; Hunt et al., 2009). The main datasets used here are the Level 2 5-km Clould Profile Products of CALIOP which provide the information of scientific parameters such as particle linear depolarization ratio ($\delta_p$), temperature (derived from the GEOS-5 data), ice water content (derived from the CALIOP retrieved
extinction by ice cloud particles), etc.

CALIOP is able to observe altitude-resolved profiles of backscatter intensity from numerous geophysical entities including clouds, aerosol layers, regions of clear air, and the returns from the Earth's surface. In this study, however, we are only interested in the cirrus ice clouds. We hence perform the vertical feature mask (VFM) developed by the CALIPSO team to distinguish cirrus clouds from other entities including aerosols as well as from non-cirrus clouds (e.g., Liu et al., 2004, 2009; Hu et
al., 2009; Omar et al., 2009; Vaughan et al., 2009). Furthermore, the archieved data of CALIOP are classified into day- and night-time. The day-time observations are affected by solar illumination, which may lead to a reduction in the signal-to-noise ratio and hence make them more difficulat to interpret. However, there is an aviation fingerprint with two maxima during eastbound and afternoon westbound traffic in the area we are focusing on here (e.g., Graf et al., 2012; Schumann and Graf, 2013). Therefore, both day- and night-time observations will be analyzed here to study the influence of air traffic on cirrus to
the fullest extent. For the detailed description of the CALIOP data, readers refer to Li and Groß (2021) and references therein.

In this study, we concentrate on the same research area as Li and Groß (2021), i.e., the midlatitude regions from 35°N to 60°N and from the Atlantic Ocean (15°W) to central Europe (15°E) (for the sake of simplicity, we call this research area as Europe in the rest of this manuscript). As a nadir-pointing lidar, CALIOP collects data only along the ground track of the CALIPSO satellite. CALIPSO flies 3-4 times each day over this area and therefore ∼100 tracks of observations each month
were collected in the years 2010-2020. Further, this area covers a large fraction of the North Atlantic flight corridor connecting central Europe with north America where the generation of contrail-induced cirrus clouds and the aviation impact on cirrus clouds have been intensively studied (e.g., Graf et al., 2012; Schumann and Graf, 2013; Voigt et al., 2017; Urbanek et al., 2018; Schumann et al., 2021a; Li and Groß, 2021).





## 3 Results

The cirrus morphologies and occurrence rates as well as the high degree of variability in their microphysical properties highly
depend on the substantial differences in meteorological conditions. Hence, we first compare the evolution of the general me-
teorological conditions along the entire altitude range from 6 to 13 km including temperature, relative humidity with respect
to ice (RHi), as well as vertical updraft and wind velocity covering our research area in years 2010–2020. These parameters
are directly derived from global ERA5 reanalysis data, produced by ECMWF with the Copernicus Climate Change Service

(Hersbach et al., 2020), and their monthly values are shown in Figure 2. The vertical bars in the upper panels stand for all the
data points of air temperature and RHi, respectively, and in the lower panels for vertical updraft and wind velocity, respec-
tively. The blue circles show the medians of each quantity in different month and the red lines are best-fitting lines using a
simple linear regression model (i.e., least squares fit of a first-degree polynomial to data) for all 4 quantities, respectively. The
derived slopes are hence considered as the long-term trends in each quantity. First of all, there is a clear seasonal cycle in the

air temperatures and higher degree of variability in temperatures can be seen in winter than in summer. Further, extreme low
temperatures with medians below -50 °C are seen in January and February 2012, which are indicative of the cold spell at early
2012 in Europe starting from 24 January and lasting for about three weeks (DWD, 2012). The influence of the extreme lower
temperatures on the properties of cirrus clouds will be discussed next. Air temperatures show in medians an increasing trend of
0.0941 °C/yr. In addition, the year-to-year variabilities of air temperatures in different month show that temperatures increased

more significantly in winter than in summer. The seasonal cycle in RHi, however, is not as significant as in air temperatures. In
general, RHi show a small decrease of medians with a trend of -0.0557%/yr, whereas their averages show an increasing trend
of 0.0777%/yr. Further, the maxima of the monthly RHi values show a decreasing trend of -0.2431%/yr, whereas their minima
an increasing trend of 0.9125%/yr. I.e., the range of the RHi distribution became narrower in the last 11 years, especially
during the period from 2010 to end 2017. Please note that the maxima of RHi play a dominant role in the contribution of the

humidity to the cirrus occurrence. It is mentioned above that the vertical updrafts play a crucial role on the formation of cirrus
clouds. However, monthly mean values of vertical updrafts determined from ERA reanalysis data are highly smoothed and can
only provide a climatological reference for the background. This is also the case for the wind velocity. Figure 2 (lower panel)
show in general a small decrease in vertical updrafts (negative values for upwards) and a small decrease in wind velocity as
well. In a nutshell, the meteorological and dynamical conditions over Europe in the last 11 years become less favorable for

cirrus formation with increasing temperature, decreasing RHi, and small variations in the dynamics. With a general picture of
meteorological conditions in mind, we can further study the seasonal variations and long-term trends in cirrus cloud occurrence
and properties.

### 3.1 Seasonal variations of cirrus clouds

We first present the distribution of $\delta_p$ of cirrus clouds in each month from January 2010 to December 2020 in Figure 3. They

are derived from the observations at the typical altitudes in which cirrus clouds form from 6 to 13 km and at temperatures
between -75 and -38 °C. Please note that typical cruising altitudes of aircrafts over the North Atlantic and Europe lie at 8.5 km



and above. However, aviation emissions over mainland take place also at lower altitudes during ascent and descent from and to the airports. In order to compare the $\delta_p$ distribution in different month, the number densities of scatter point data are normalized and visualized with different color codes with the maximum number density indicated by 1 for each individual month. February

2016 with no observations available is marked in blank. Figure 3 provides a general climatology of the distributions of $\delta_p$ in Europe with the majority of $\delta_p$ (> 60% of the maxima) falling within the range from 0.2 to 0.55. There are clear seasonal cycles as expected in $\delta_p$, which becomes more remarkable for them falling within the smaller range of $\delta_p$. In addition, there are clear reductions in $\delta_p$ during the periods of COVID-19 from March to December 2020 compared to the corresponding months in the previous years, which is consistent with the previous studies (Li and Groß, 2021; Voigt et al., 2022) and will be discussed

in details below. Furthermore, the reductions in the cirrus $\delta_p$ are more remarkable for the measurements in the day time. The occurrence rates (OR) of cirrus cloud during the period of COVID also show a reduction, which has been reported recently (e.g., Schumann et al., 2021a; Quaas et al., 2021; Li and Groß, 2021).

We next focus on the seasonality in cirrus cloud occurrence and $\delta_p$ in more details. In Figure 4, we show the cirrus OR and $\delta_p$ in different month (see the legend on the plot with descriptive labels) for each 1-km altitude bin from 6 to 13 km.

The data with cirrus OR smaller than 0.1% are neglected for plotting $\delta_p$ in the right panel. The profiles of cirrus OR along altitudes show that cirrus clouds mainly occurred in the height range from 9 to 11 km (with OR > 3% in every month). The cirrus occurrence follows a significant seasonal cycle in all the altitude bins. The maximum of cirrus OR, up to 11%, are found in the winter months, more precisely, in January or February below 10 km and in December above, respectively, and the minimum OR appear in July along altitudes. In addition, there is a stronger seasonality in the lower altitudes with the cirrus

OR in winter more than 10 times larger than those in summer. Please note that excluding the measurements of cirrus due to deep convection leads to a reduction in the cirrus occurrence rate, especially in summer. Further, cirrus clouds in the winter months appear within the full altitude range from 6 to 13 km while in summer only from 9 to 12.5 km for the data with OR > 1% considered. For the $\delta_p$ values of cirrus shown in the right panel, they also follow a distinct seasonal cycle that cirrus clouds are characterized by larger $\delta_p$ values in winter than in summer in each altitude bin and the difference of $\delta_p$ in different months

can be as large as 0.06 (~15%). The distributions of cirrus $\delta_p$ along altitudes show a clear increase with altitudes in each month and the difference of the medians of $\delta_p$ in each month can be as large as more than 0.1 (Urbanek et al., 2018; Li and Groß, 2021).

### 3.2  Long-term trend of cirrus cloud properties with significance test

In Figure 5 we present the $\delta_p$ medians of cirrus clouds in every month derived from both day- and night-time observations

(in the upper panel) and from only the day-time observations (in the lower panel), respectively. First of all, the $\delta_p$ values determined from the day-time observations are generally larger than those from combined day- and night-time observations by ~0.04 on average. For both cases, we notice that the cirrus $\delta_p$ shows clear reductions in the period of COVID-19 pandemic starting from March 2020 (shown with squares in gray in Figure 5) and the reductions are stronger in $\delta_p$ derived from the day-time observations. The same findings on the changes in the cirrus cloud properties and occurrence in March and April

2020 compared to the previous years have been reported by Li and Groß (2021) as well as in early summer 2020 during



the BLUESKY campaign (Voigt et al., 2022). Hence, in this study only observations before the COVID-19 pandemic are considered for further analysis. In order to calculate the long-term trends in the cirrus $\delta_p$, we apply two methods in this study, i.e., ordinary least square (OLS) estimator and Theil-Sen estimator (TSE). The OLS estimator is a commonly-used method to estimate the unknown parameters in a linear regression model by minimizing the sum of the squares of the differences

between the observed dependent variables (here $\delta_p$) and those predicted by the linear function of the independent variable (time in month). However, OLS is highly affected by the presence of outliers in the time series making the estimation less efficient. The TSE method is a nonparametric estimation technique by calculating all the slopes between pairs of points and choosing the median as the estimation of the regression slope. Compared to OLS, TSE is a robust linear regression against outliers since it uses medians instead of means. The calculated long-term trends (i.e., slopes) and the regressed linear fits with

both methods are shown on the plot. It is mentioned above that the aviation densities in Europe grew more strongly since 2013 (see Figure 1). We hence further calculate the trends for the observations during a shorter period from March 2013 to February 2020 and the corresponding results are also indicated in Figure 5. First of all, $\delta_p$ shows an increasing trend with a slope of 0.77e-3/yr and 1.02e-3/yr with both methods, respectively, determined from all the observations (including day and night) in the last 10 years before COVID. We notice the exceptions that cirrus clouds are characterized by extremely large values of

$\delta_p$ in January and February 2012, which might be connected with the cold spell in Europe early 2012 (e.g., DWD, 2012). We further compare the occurrence heights of cirrus clouds in January and February 2012 with other years and find that they in distribution are higher than those in other years. It is the other way around that the extremely low values of $\delta_p$ in July 2012 are correlated with the much lower occurrence heights of cirrus clouds in July 2012 than in other years. The interpretations for the results are based on the altitude dependence of the cirrus $\delta_p$ that has been presented in Figure 4 and also reported in

previous studies (e.g., Urbanek et al., 2018; Li and Groß, 2021). For the further analysis the extreme values are considered as outliers and removed. The interpolated data are then conducted with both OLS and TSE methods and the derived slopes are 0.87e-3/yr and 0.97e-3/yr, respectively. These exercises imply that the TSE method is more efficient for the here-analyzed data. Comparing with the changes in aviation, we find that $\delta_p$ generally increases following the increasing aviation densities. This is consistent with the previous studies showing that cirrus clouds with enhanced $\delta_p$ values form in areas of high aviation

emissions or vice versa (e.g., Urbanek et al., 2018; Li and Groß, 2021). Furthermore, larger trends are expected and derived as well from the observations in a shorter period from March 2013 with a slope of 1.18e-3/yr and 1.51e-3/yr with both methods, respectively. The same analyses are extended to the day-time observations and the corresponding results (see the lower panel of Figure 5) show slightly larger trends of 0.93e-3/yr and 1.09/yr with both OLS and TSE, respectively, than the results derived from the combined day- and night-time observations. All the results of long-term trends with the medians of total $\delta_p$ values are

summarized in Table 1 and 2. The comparison indicates that aviation exerts stronger impacts on cirrus clouds in the day time than in night since there is an aviation fingerprint with two maxima during morning eastbound and afternoon westbound traffic in the north Atlantic flight corridor covering the area in this study (e.g., Graf et al., 2012; Schumann and Graf, 2013).

It is mentioned in the previous subsection that cirrus cloud properties ($\delta_p$) and occurrence rates are dominated by seasonal cycles. Spectral analysis with Fourier transform is carried out on the time series of $\delta_p$ (not shown here) and the periodogram

of $\delta_p$ with a dominant peak of power at the point of 12-month cycle indicates a conspicuous seasonality (aka annual cycle).





This repeating cycle of seasonality may obscure the long-term trend in the data that we want to determine. We therefore deseasonalize the data by computing the monthly climatological mean, subtracting them from each monthly record and finally adding the total mean of $\delta_p$. The deseasonalized $\delta_p$ values are shown in Figure 6: upper panel for the combined day- and night-time observations and lower panel for only the day-time observations. The detection of seasonal anomalies of extremely large $\delta_p$

values in winter and extremely small $\delta_p$ values in summer are easy, but the opponent cases are not. With the deseasonalization process, an outlier of February 2018 is detected which, however, will not bias the calculation by using the TSE method. The results of long-term trends again calculated with both OLS and TSE methods and the regressed linear fits are shown in Figure 6. The deseasonalized $\delta_p$ values show a long-term trend of 0.67e-3/yr (with TSE) and 1.02e-3/yr (with TSE) for the day- and night-time observations as well as for only the day-time observations, respectively.

The significance tests of the derived trends in the cirrus $\delta_p$ values and the deseasonalized values of them are carried out applying the Mann-Kendall (MK) test (Mann, 1945; Kendall, 1975). It is a rank-based nonparametric method that has been widely used to statistically assess whether there is a monotonic trend in a time series of environmental and hydrological data (e.g., Yue et al., 2002) (see Appendix B). The overall results of the MK test for the long-term trends in $\delta_p$ at a significance level of $p = 5\%$ are presented in Tables 1 and 2 for all the observations (including day- and night-time) and for only the day-time

observations, respectively. Here, the $p$ value returned from a MK test is a measure of the probability of rejecting or retaining the null hypothesis H0 stating that the data are independently distributed with no trend. $h$ is logical value (0 or 1) used to give the test decision: $h = 1$ indicates a rejection of the null hypothesis (i.e., no trend) and $h = 0$ indicates a failure to reject it at the 5% significance level. From the results of the MK test, it is striking that significant increasing trends (all with $h = 1$) exist in the monthly values of $\delta_p$ and their deseasonalized values in the full period from March 2010 to February 2020 as well as in a

shorter period from March 2013 to February 2020.

    We next turn to the determination of trend in the occurrence rates of cirrus clouds. Again, we focus on the observations of cirrus clouds at the altitudes from 6 to 13 km and at the temperatures from -75 to -38°C. The corresponding results are shown in Figure 7 (upper panel). Clearly, a strong seasonal cycle also exists in the occurrence rate with a monthly mean value up to 8% in the winter months and as low as 1% in summer. Following the same procedure, the deseasonalization of cirrus

occurrence rate has been carried out and the corresponding results are shown in the lower panel of Figure 7. The long-term trends are calculated with the TSE method and the linear fits are overplotted in Figure 7 showing negative values. I.e., the cirrus occurrence rates decrease with time in the last 10 years before COVID. However, the significance tests with the MK test show that only a significant trend of -0.0916%/yr exists in the deseasonalized OR for the period from March 2013 to February 2020. Nevertheless, the decreasing OR of cirrus can be traced back to the changes in the meteorological conditions, i.e., with

increasing temperatures and decreasing humidity, in the period of interest (see Figure 2).

    It is presented above that $\delta_p$ generally increases with altitudes. We hence determine the occurrence heights of cirrus clouds and their monthly means are shown in Figure 8. The occurrence heights of cirrus also follow a distinct seasonal cycle with larger means in the summer months and smaller values in winter which indicates that cirrus clouds in winter occur at the full altitude range from 6 to 13 km but in summer only at higher altitudes. The same procedures to calculate the long-term trends

with the TSE method have been carried out and the corresponding results are shown in Figure 8. It is striking that the occurrence



**Table 1.** Long-term trends of monthly values of particle linear depolarization ratio (PLDR) $\delta_p$ within the typical cirrus altitude range from 6 to 13 km derived from both day- and night-time data during the period from Mar. 2010 to Feb. 2020: Comparison between two methods of the ordinary least square (OLS) estimator and the Theil-Sen estimator (TSE), including Mann-Kendall (MK) significance test.

| | Median | OR of cirrus (%) | Trend (/yr, OLS) | Trend (/yr, TSE) | $h$ | $p$ value |
|---|---|---|---|---|---|---|
| $\delta_p$ (03.2010-02.2020) | 0.3701 | 4.3679 | 0.7732e-3 | 1.0170e-3 | 1 | 0.0079 |
| Deseasonalized $\delta_p$ (03.2010-02.2020) | | | 0.4703e-3 | 0.6676e-3 | 1 | < 0.001 |
| $\delta_p$ (03.2013-02.2020) | 0.3714 | 4.4451 | 1.1754e-3 | 1.5084e-3 | 1 | 0.0258 |
| Deseasonalized $\delta_p$ (03.2013-02.2020) | | | 0.5572e-3 | 0.7197e-3 | 1 | < 0.001 |

**Table 2.** Same as Table 1, but for the results derived from only day-time data during the period from Mar. 2010 to Feb. 2020.

| | Median | OR of cirrus (%) | Trend (/yr, OLS) | Trend (/yr, TSE) | $h$ | $p$ value |
|---|---|---|---|---|---|---|
| $\delta_p$ (03.2010-02.2020) | 0.4084 | 3.6335 | 0.9301e-3 | 1.0946e-3 | 1 | < 0.001 |
| Deseasonalized $\delta_p$ (03.2010-02.2020) | | | 0.8325e-3 | 1.0210e-3 | 1 | < 0.001 |
| $\delta_p$ (03.2013-02.2020) | 0.4106 | 3,6335 | 1.1073e-3 | 1.3548e-3 | 1 | < 0.001 |
| Deseasonalized $\delta_p$ (03.2013-02.2020) | | | 0.9082e-3 | 1.0483e-3 | 1 | < 0.001 |

heights of cirrus cloud show a trend of 0.0238 km/yr (0.0447 km/yr) during the period from March 2010 to February 2020 (from March 2013 to February 2020). After deseasonalizing the data, we reach to a long-term trend of 0.0309 km/yr (0.0508 km/yr) for the full period (the shorter period). Importantly, the trends in the deseasonalized values of cirrus occurrence heights are statistically significant according to the results with the MK test. These findings are consistent with the upward shift of the aircraft cruising altitudes in the last years due to the fact that aircrafts flying higher leads to less fuel burn and hence less CO2 emissions (https://www.eurocontrol.int/publication/eurocontrol-data-snapshot-21-aircraft-flying-higher-more-efficiently-and-sustainably, last access: 13 April 2022). In contrast to this, however, an upward displacement of air traffics may lead to the increase of the contrail coverage (e.g., Fichter et al., 2005).

## 4 Discussion

### 4.1 Correlation with the ambient temperatures

It is mentioned that temperatures and other meteorological parameters play a decisive role in the formation and maintenance of cirrus clouds (e.g., Bailey and Hallett, 2004; Um et al., 2015). Given the scope of this study, we will only discuss the contributions of temperature to the cirrus cloud properties. We first compare the relationship between the cirrus $\delta_p$ and the corresponding ambient temperatures. The temperatures used for this analysis are derived from the GEOS-5 (Goddard Earth



**Table 3.** Long-term trends of monthly occurrence rate (OR) of cirrus clouds at the typical cirrus altitude range from 6 to 13 km derived from both day- and night-time data during the period from Mar. 2010 to Feb. 2020 with the Theil-Sen estimator (TSE) method including Mann-Kendall (MK) significance test.

|  | Median (%) | Trend (%/yr, TSE) | $h$ | $p$ value | $Z_{MK}$ |
|---|---|---|---|---|---|
| OR (03.2010-02.2020) | 4.2660 | -0.0017 | 0 | 0.9910 | -0.0113 |
| Deseasonalized OR (03.2010-02.2020) | 4.4578 | -0.0121 | 0 | 0.4466 | -0.7610 |
| OR (03.2013-02.2020) | 4.3701 | -0.0795 | 0 | 0.1962 | -1.2923 |
| Deseasonalized OR (03.2013-02.2020) | 4.4515 | -0.0916 | 1 | 0.0098 | -2.5844 |

**Table 4.** Long-term trends of monthly occurrence height (OH) of cirrus clouds at the typical cirrus altitude range from 6 to 13 km derived from both day- and night-time data during the period from Mar. 2010 to Feb. 2020 with the Theil-Sen estimator (TSE) method including Mann-Kendall (MK) significance test.

|  | Median (km) | Trend (km/yr, TSE) | $h$ | $p$ value | $Z_{MK}$ |
|---|---|---|---|---|---|
| OH (03.2010-02.2020) | 10.06 | 0.0238 | 0 | 0.1202 | 1.5541 |
| Deseasonalized OH (03.2010-02.2020) | 9.95 | 0.0309 | 0 | 0.1022 | 1.6345 |
| OH (03.2013-02.2020) | 10.06 | 0.0447 | 1 | < 0.001 | 4.2646 |
| Deseasonalized OH (03.2013-02.2020) | 9.96 | 0.0508 | 1 | < 0.001 | 5.1694 |

Observing System, version 5) model data product provided to the CALIPSO by the GMAO (Global Modeling and Assimilation Office) data assimilation system. In Figure 9, we present a general picture of the determined relationship between both quantities with a heatmap to specify the number density of the scatter point data. The dependence of the cirrus $\delta_p$ on the ambient temperatures shows clear different variations separated at a temperature threshold of $\sim$ -50 °C. At temperatures below -50 °C, the $\delta_p$ values are roughly negatively correlated with the ambient temperatures. At temperatures warmer than -50 °C, however,

$\delta_p$ with the majority of values falling within the range from $\sim$0.23 to 0.45 shows no clear correlation with the ambient temperatures. The same relationship was determined also for the monthly data in April from 2014 to 2020 by Li and Groß (2021). For all the data in total, the $\delta_p$ values show a negative correlation with the corresponding ambient temperatures.

We further calculate the medians of $\delta_p$ of cirrus clouds in each month from March 2010 to February 2020 as well as the medians of ambient temperatures in cirrus clouds and the corresponding results are presented in the left panel of Figure 10. The

scatter plots clearly shows that the cirrus $\delta_p$ values increase following the decreasing ambient temperatures with a correlation coefficient $r = -0.76$, which is statistically significant ($p < 0.0001$). Due to the strong influence of the ambient temperatures on the properties of cirrus clouds, an appropriate method is needed to eliminate the induced parts in cirrus $\delta_p$ by temperatures. Generally, regression models that may be applied to isolate the temperature-induced parts fall into two kinds, i.e., based on either linear or nonlinear dependence (quadratic or cubic):





$$X_{\text{th}} = b_0 + b_1 \cdot T \tag{2}$$

$$X_{\text{th}} = b_0 + b_1 \cdot T + b_2 \cdot T^2 \tag{3}$$

$$X_{\text{th}} = b_0 + b_1 \cdot T + b_2 \cdot T^2 + b_3 \cdot T^3 \tag{4}$$

where the variable $T$ stands for the ambient temperature and $X_{\text{th}}$ for the theoretical value of $\delta_p$ regressed from temperatures. After regression analyses, the absolute deviation $\Delta X$ (i.e., the residual) of the observational data $X_{\text{obs}}$ from the corresponding $X_{\text{th}}$ can be calculated by

$$\Delta X = X_{\text{obs}} - X_{\text{th}} \tag{5}$$

Using the derived $\Delta X$, linear trends can further be determined using a simple linear regression model:

$$\Delta X = d + m \cdot \text{time} \tag{6}$$

where "time" stands for each month, season, or year for which data are available. For this study, a linear trend coefficient (slope) $m$ can be determined from the monthly values of $\delta_p$ medians using Equation (6).

Applying the linear dependence by Equation (2) as well as Equation (6), we determine the regressed $\delta_p$ values and the residuals (see the right panels of Figure 10). For a comparison, we also determine the regressed values according to the quadratic and cubic dependence with Equations (3) and (4) which, however, show no big difference from the results based on the linear dependence. The long-term trend in the residuals is calculated with the TSE method showing an increase of 0.54e-3/yr which is above 90% confidence level according to the MK test with $p = 0.0582$. The same analyses have been also carried out to the day-time observations resulting in a long-term trend of 0.69e-3/yr in the residuals of $\delta_p$ which is statistically significant with $p < 0.001$ (see Figure 11). We should note that the dependence of the cirrus $\delta_p$ on the ambient temperatures to the full extent is not linear but shows different characteristics in different temperature range (roughly separated at ∼-50 °C shown in Figure 9). However, the monthly mean values of both quantities along the full altitudes of cirrus clouds can still provide a climatological relationship between them. After removing the temperature-induced parts in the cirrus $\delta_p$, the residuals still show an increasing trend over this period, which is supposed to be due to other factors than the meteorological conditions.

It is discussed above that the occurrence heights of cirrus clouds show an increase on average over this period which, in general, leads to a decrease of temperatures in clouds. The altitude dependence of temperatures is assumed based on the fact that the tropopause heights are roughly 1.2 km higher than the peak of the cirrus occurrence heights (i.e., the most probable altitudes at which cirrus clouds occurred) and both parameters are highly correlated including their seasonal cycles and trends.





## 4.2 Correlation with the air traffic densities over Europe

Using the airborne lidar measurements during the ML-CIRRUS campaign in 2014 (Voigt et al., 2017), Urbanek et al. (2018)
concentrated on the specific clouds and found lower supersaturation in the cirrus clouds with enhanced $\delta_p$ which were traced
back to be forming in areas of high aviation emissions. Recently, Li and Groß (2021) analyzed the satellite data of CALIPSO
which cover the European area (i.e., the same research area as the current study) and found that strong reductions in air traffic
in Europe caused by the COVID-19 pandemic led to significant changes in cirrus cloud properties in terms of $\delta_p$. In this study,
we further study the impact of the increasing air traffic on the cirrus clouds in a longer period of 10 years before COVID. It is
described above that the air traffic densities in the 42 European countries and regions (see Figure 1) show a clear seasonal cycle
which is, however, roughly anticorrelated with the seasonality in the cirrus $\delta_p$. To directly compare them, we first deseasonalize
both datasets (here considering CO2 emissions as a proxy of air traffic densities) and the results are shown in Figure 12. Please
note that both number of flights and CO2 emissions from aviation are available and analyzed here. The results derived from
both datasets are consistent. Hence, we will only concentrate on the influence of CO2 emissions from aviation. According to
the description above, the outliers in the deseasonalized $\delta_p$ have been removed and further interpolated (see the upper panel
of Figure 6). Furthermore, the annual mean variations of the deseasonalized $\delta_p$ are derived using a 12-point moving average
smooth (shown in red in Figure 12). Generally, an increasing trend can be recognized in both parameters, especially in the
period from January 2013 to February 2020. The correlation coefficients between them for a full period from January 2010
to February 2020 are calculated showing $r = 0.25$ and $r = 0.54$, respectively. Larger values of the correlation coefficients are
also derived for a shorter period from January 2013 to February 2020. The confidence levels for all the cases are above 99.5%.

With the general picture between the cirrus $\delta_p$ and air traffic in the 10-year period in mind, we may further focus on their
correlations in different seasons. To illustrate the seasonal variations of the relationships between the cirrus $\delta_p$ and air traffic,
we show the comparison between both parameters in different seasons (winter: December-February, but only January and
February for 2010; spring: March-May; summer: Jun-August; autumn: September-November) in Figure 13. First of all, the
cirrus $\delta_p$ values in all the seasons show increasing trends in the last 10 years before COVID. The trend in summer, however,
is much smaller (10 times smaller) than those in other seasons. Strong reductions in air traffic caused by COVID starting
from March 2020 lead to corresponding reductions in the cirrus $\delta_p$ in the spring, summer, and autumn in 2020, but a slight
increase in the winter 2020 (including December 2019) (Li and Groß, 2021). The calculated correlation coefficients between
both parameters are all positive, but the confidence levels are above 95% only for summer and autumn. We also calculate the
correlation coefficients excluding the COVID period, i.e., for winter from 2010 to 2020 and for other seasons from 2010 to
2019, showing the confidence levels above 95% only for autumn. Finally, we perform the same calculations of correlation
coefficients on the monthly $\delta_p$ and corresponding CO2 emissions from aviation in different seasons excluding the COVID
period and the results along with the corresponding $p$-values at the confidence level of 95% are listed in Table 5.



**Table 5.** Correlation coefficients between the monthly $\delta_p$ of cirrus clouds within the altitude from 6 to 13 km and corresponding CO2 emissions from aviation as well as number of flights in Europe in different seasons.

| Season | Winter (DJF) | Spring (MAM) | Summer (JJA) | Autumn (SON) |
|---|---|---|---|---|
| CO2 emission (03.2010-02.2020) | r = 0.18, p = 0.3211 | r = 0.30, p = 0.1345 | r = 0.16, p = 0.4247 | r = 0.49, p = 0.0098 |
| CO2 emission (03.2013-02.2020) | r = 0.57, p = 0.0047 | r = 0.46, p= 0.0354 | r = 0.05, p = 0.8216 | r = 0.61, p = 0.0034 |
| Number of flights (03.2010-02.2020) | r = 0.25, p = 0.1624 | r = 0.37, p = 0.0573 | r = 0.11, p = 0.5802 | r = 0.40, p = 0.0373 |
| Number of flights (03.2013-02.2020) | r = 0.49, p = 0.0170 | r = 0.60, p = 0.0039 | r = 0.02, p = 0.9376 | r = 0.53, p = 0.0125 |

## 5 Conclusions

Motivated by the work by Li and Groß (2021) who presented the changes in cirrus cloud properties and occurrence caused by the reduced air traffic during the COVID-19 pandemic, we carried out in the current study further analyses of 10-year lidar measurements of cirrus clouds with CALIPSO before COVID. Over this period, aviation grew strongly in terms of CO2 emissions and flight densities in Europe, especially from 2013 to early 2020. The meteorological conditions, however, became less favorable for cirrus formation with increasing temperatures and decreasing humidity.

The results show that cirrus clouds follow a distinct seasonal cycle in their appearance including occurrence rate (OR) and occurrence height (OH) as well as in the particle linear depolarization ratio (PLDR) $\delta_p$. Cirrus clouds in the winter months occurred within a broader altitude range from 6 to 13 km than in summer (only from 9 to 12.5 km) and their OR in winter can be more than 10 times larger than those in summer. Further, the seasonal cycles in cirrus OR are recognized along the whole altitudes where cirrus clouds form and seem to be more remarkable in the lower altitudes. The $\delta_p$ values of cirrus show the 360 majority of the data falling within the range from 0.2 to 0.55. Cirrus clouds are characterized by larger values of $\delta_p$ in winter than in summer. In addition, $\delta_p$ shows generally a clear increase along the altitudes in each month, which is consistent with previous studies (e.g., Urbanek et al., 2018; Li and Groß, 2021).

The medians of $\delta_p$ in each month were first calculated from the measurements from 2010 to 2020 and they show a significant reduction during the period of COVID starting from March 2020 which is supposed to be caused by the reduced air traffic (Li 365 and Groß, 2021; Voigt et al., 2022). The long-term evolution of the cirrus $\delta_p$ before COVID shows an increasing trend with a slope of 1.02e-3/yr (1.51e-3/yr) in the period from March 2010 to February 2020 (from March 2013 to February 2020) based on the calculation with the Theil-Sen Estimator (TSE) method. The derived trends are both above a confidence level of 95% according to the Mann-Kendall (MK) significance test. The long-term trend of $\delta_p$ (a slope of 1.09e-3/yr) was also derived from the day-time observations showing a slightly larger value than the result determined from both day- and night-time 370 observations. Since the cirrus cloud occurrence and $\delta_p$ are dominated by seasonal cycles, we further deseasonalized the time series of monthly medians of $\delta_p$. The deseasonalized $\delta_p$ values show a long-term trend of 0.67e-3/yr (with TSE) at a confidence level of 99.5%. The cirrus occurrence rates as well as the deseasonalized values, however, both show a small negative trend over this period, which is supposed to be connected with the background meteorological conditions. Furthermore, there are a



strong increasing trend in the cirrus occurrence heights as well as in the deseasonalized data, which is very striking since the
findings correspond to the upward shift of the aircraft cruising altitudes in the last years.

To study the potential reasons for the detected trends in cirrus clouds, we first compare the relationship between the cirrus $\delta_p$ and the corresponding ambient temperatures from all the data, which show a negative correlation at temperatures below -50 °C and no clear correlation above. For the monthly medians, however, a significant linear correlation was reached between both parameters with a correlation coefficient of -0.76. We hence regressed $\delta_p$ with a simple linear regression model because
of the strong dependence of $\delta_p$ on the ambient temperatures and removed the temperature-induced contributions from the cirrus $\delta_p$. The derived residuals reveal an increasing trend of 0.54e-3/yr above 90% confidence level according to the MK test, which should be induced by other factors than temperatures. Before we carry out the comparison between the cirrus $\delta_p$ and the corresponding aviation, we should first deseasonalize both datasets since they follow totally different seasonal cycles (roughly anticorrelated). In general, there is a conspicuous increasing trend in the deseasonalized time series of $\delta_p$ as well
as the corresponding CO2 emissions from aviation, especially in the period from January 2013 to February 2020. The close correlation between them shows a correlation coefficient of 0.25 (0.58) in the full period from January 2010 to February 2020 (from January 2013 to February 2020), which are at the confidence level above 99.5%. We further compared the relationship between the cirrus $\delta_p$ and the corresponding CO2 emissions from aviation in different season. Concentrating on the data before the period of COVID, we calculated the correlation coefficients between both parameters based on their monthly values in
different seasons and reached a strong correlation in winter, spring, and autumn, but a weak correlation in summer.

*Code availability.*  Data description and example codes for handling the VFM data are available at https://www-calipso.larc.nasa.gov/ (NASA, 2021a). The MATLAB codes for drawing the plots in this paper can be made available upon request.

*Data availability.*  The CALIPSO data, including VFM used in this study, can be obtained via https://subset.larc.nasa.gov/calipso/login.php (NASA, 2021b, login required). ECMWF ERA5 data can be freely accessed from https://www.ecmwf.int/en/forecasts/datasets/reanalysis-
datasets/era5. The reanalyzed data of cirrus parameters can be made available upon request.

*Author contributions.*  QL collected and analyzed the data and wrote the manuscript with help from SG. Both authors discussed the results and findings and contributed to finalizing the manuscript.

*Competing interests.*  The authors declare that they have no conflict of interest.





**Appendix A: Ordinary least squares (OLS) estimator and Theil-Sen estimator (TSE)**

The ordinary least square (OLS) estimator is a common technique for estimating coefficients of linear regression equations which describe the relationship between one or more independent quantitative variables and a dependent variable (simple or multiple linear regression). The OLS estimator chooses the coefficients by minimizing the sum of the squares of the differences between the observed dependent variables and the predicted values by a linear regression functions of the independent variable. The OLS estimator is highly biased by the outliers in the time series. Presence of outliers departs the distribution of errors away

from a normal distribution resulting in heavy tails due to greater standard error than the expected.

The Theil-Sen estimator (or Theil-Sen regression or Sen tau method) is a nonparametric estimation technique for estimating a linear trend (i.e., only for one-variable regression), which uses median instead of mean. Hence, this estimator is not sensitive to outliers. The idea behind the estimator is simple. The slopes $S$ between all pairwise sets of observations are computed and the medians of all these slopes are chosen as the estimate of the regression slope.

The simple linear regression model based on the observations $(x_i, y_i)$, is $y_i = \alpha + \beta x_i + \epsilon_i$, i = 1,2,...,n. Where $y$ is the dependent variable, $x$ is the independent variable, $\alpha$ and $\beta$ are intercept and slope parameters, respectively, and $\epsilon$ is the error term. The Theil-Sen slope estimate is obtained by taking the median of all

$$s_{ij} = \frac{y_j - y_i}{x_j - x_i}, \tag{A1}$$

where $1 \leq i \leq j \leq n$. The intercept $b$ is obtained by taking the median of all the differences $(y_i - s_i x_i)$.

**Appendix B: Mann-Kendall test**

The Mann-Kendall (MK) trend test (Mann, 1945; Kendall, 1975) is a nonparametric test (i.e., no underlying assumption made about the distribution of the data) widely used to statistically assess whether there is a monotonic increasing or decreasing trend in a time series of climatologic and hydrologic data, even if there is a seasonal component in the time series. The null hypothesis, H0, states that the data are independently distributed with no trend. The alternative hypothesis, H1, is that the data

follow a monotonic trend. According to the test, each data value of the time series with $n$ data points is compared with all subsequent data values. The statistic $S$ is incremented by 1, if a data value from a later time period is higher than a data value sampled earlier, otherwise, $S$ is decremented by 1. I.e., the MK test statistic $S$ is calculated by:

$$S = \sum_{i=1}^{n-1} \sum_{j=i+1}^{n} \text{sign}(X_j - X_i) \tag{B1}$$

where $X_i$ and $X_j$ are the values of sequence $i$, $j$; $n$ is the length of the time series and

$$\text{sign}(x) = \begin{cases} 1, & \text{if } x > 0 \\ 0, & \text{if } x = 0 \\ -1, & \text{if } x < 0 \end{cases} \tag{B2}$$



Mann (1945) and Kendall (1975) have documented that the statistic $S$ is approximately normally distributed when $n \geq 8$. The mean of $S$ is $E(S) = 0$ and the variance $\sigma^2$ of $S$ is defined by:

$$\sigma^2 = \frac{n(n-1)(2n+5) - \sum_{j=1}^{m} T_j(T_j - 1)(2T_j + 5)}{18} \tag{B3}$$

where $m$ is the number of the groups of tied ranks and $T_j$ is the number of data points in the $j$th tied group. The standardized
test statistic $Z_{\mathrm{MK}}$ is computed by

$$Z_{\mathrm{MK}} = \begin{cases} \frac{S-1}{\sigma}, & \text{if } S > 0 \\ 0, & \text{if } S = 0 \\ \frac{S+1}{\sigma}, & \text{if } S < 0 \end{cases} \tag{B4}$$

A positive (negative) value of the statistic $Z_{\mathrm{MK}}$ indicates that the data show an increase (decrease) with time. Given a confidence level $\alpha$, the null hypothesis is rejected (i.e., exist of a statistically significant trend in the time series) if the absolute value of $Z_{\mathrm{MK}}$ is larger than the theoretical value $Z_{1-\alpha/2}$ (for two-tailed test) or $Z_{1-\alpha}$ (for one-tailed test), indicating that a
significant trend exists in the time series. Here $Z_{1-\alpha/2}$ denotes the 100(1-$\alpha$/2)th percentile of the standard normal distribution and can be obtained from the standard normal $Z$-table. In this study a confidence level $\alpha = 0.05$ is used. At the significance level $p = 5\%$, the null hypothesis is rejected if $|Z_{\mathrm{MK}}| > 1.96$.

*Acknowledgements.* The authors acknowledge the financial support by DLR internal funding within the MABAK project (Innovative Methoden zur Analyse und Bewertung von Veränderungen der Atmosphäre und des Klimasystems). We thank the NASA Langley Research
Center Atmospheric Science Data Center (ASDC) and CALIPSO science team for making the data available for research. Furthermore, we acknowledge ECMWF for providing the ERA5-data from the Copernicus Climate Change Service (C3S) Climate Data Store.



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




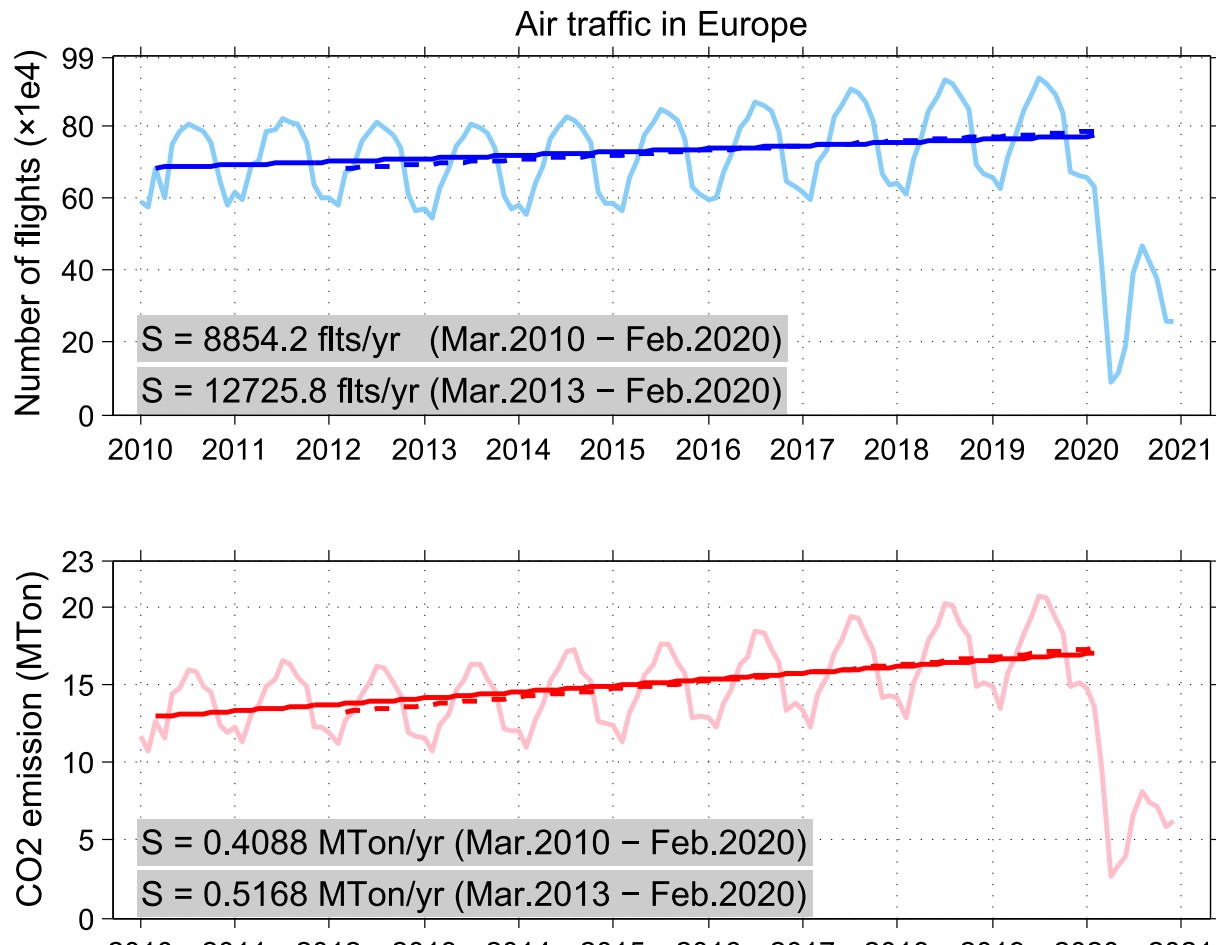

**Figure 1.** Number of flights and the corresponding CO2 emissions from air traffic in different month over Europe (in total 42 countries and regions) in years from 2010 to 2020. The seasonal cycle in the air traffic shows more flights (as well as CO2 emissions) in summer than in winter. The number of flights increased by about 1.30% /yr, whereas the CO2 emissions increased by about 3.16%/ year in the last 10 years in Europe before the COVID-19. During the COVID-19 pandemic, however, civil aviation in Europe was significantly reduced since March 2020 with only a partial recovery in summer 2020 and further reduction afterwards. Furthermore, we also note that there was actually a slightly decrease in the first three years from 2010 to 2013, especially in winter and early spring. The plots are reproduced based on the European flight historic data from the European Organisation for the Safety of Air Navigation (EUROCONTROL, https://www.eurocontrol.int/covid19, last access: 09 February 2022).





**Figure 2.** Monthly variations of air temperature, relative humidity with respect to ice (RHi), vertical updraft, and wind velocity in the background at altitudes from 6 to 13 km, derived from ERA5 reanalysis data over the European area (i.e., the research area focused in this study, Lat: 35 to 60 °N, Lon: 15°W to 15°E). The vertical bars show all the data point for temperatures and RHi, respectively, in the upper panel and for vertical updraft and wind, respectively, in the lower panel. The blue circles show the medians of each quantity in different month and the red lines are the best-fitting lines using the simple linear regression to data for all of them.





**Figure 3.** Number densities of the cirrus $\delta_p$ distribution normalized for each month from Jan 2010 to Dec 2020. The data are derived from the observations at the typical cirrus heights from 6 to 13 km and at temperatures between -75 and -38 °C. The color codes are used to visualize the relative number densities of scatter point data, with the maximum number density indicated by 1 in the color bar.

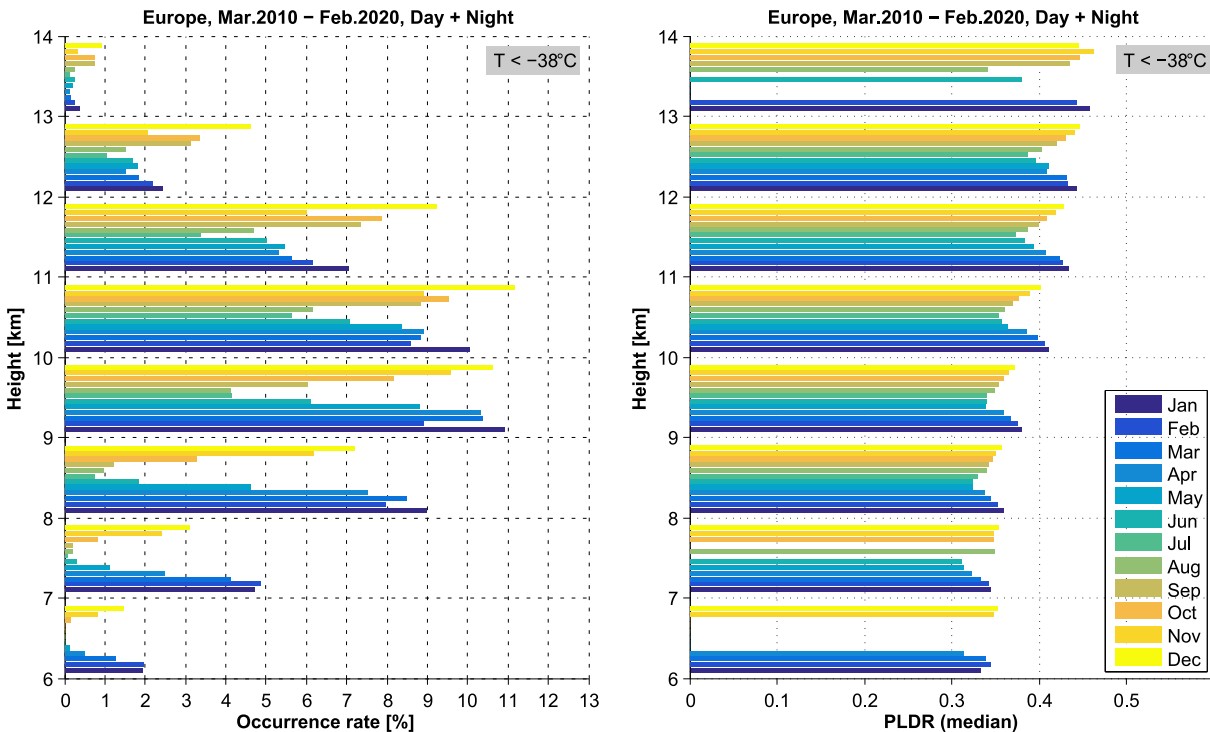

**Figure 4.** Distributions of the occurrence rate (OR) and $\delta_p$ of cirrus clouds in each 1-km altitude bin. The data are derived from the full altitude range of cirrus at temperatures between -75 and -38 °C. The data with OR < 0.1% were neglected for plotting $\delta_p$ in the right panel. Both parameters follow a seasonal cycle: cirrus clouds in winter are characterized by larger values of $\delta_p$ and higher occurrence rates than in summer. Furthermore, cirrus clouds appear within the full altitude range from 6 to 13 km in the winter months but only within the altitudes from ∼9 to 12.5 km in summer.





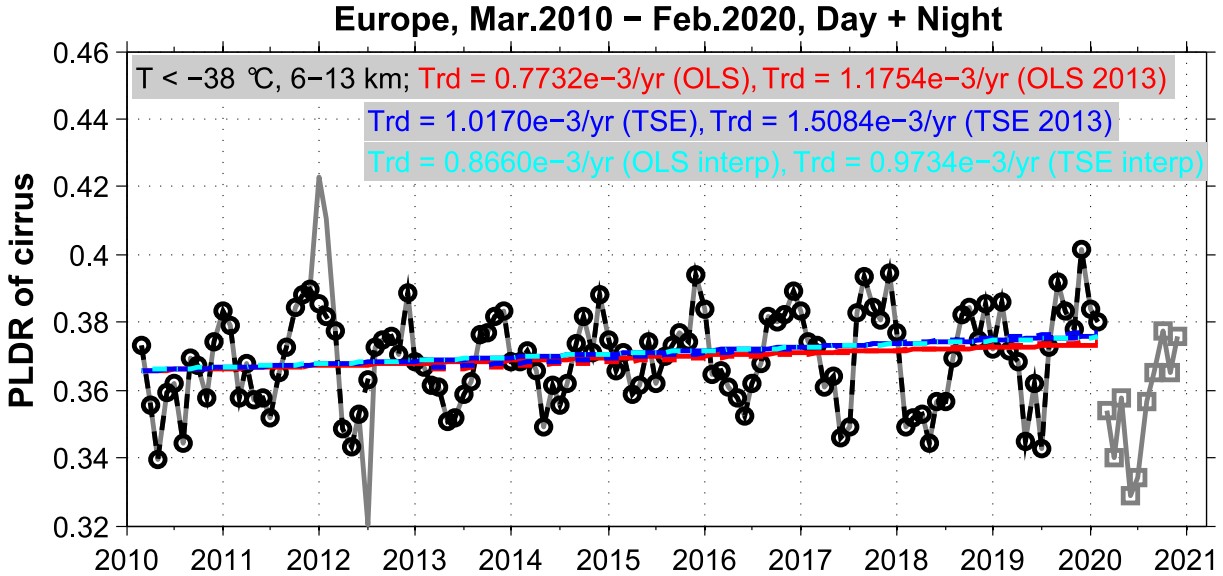

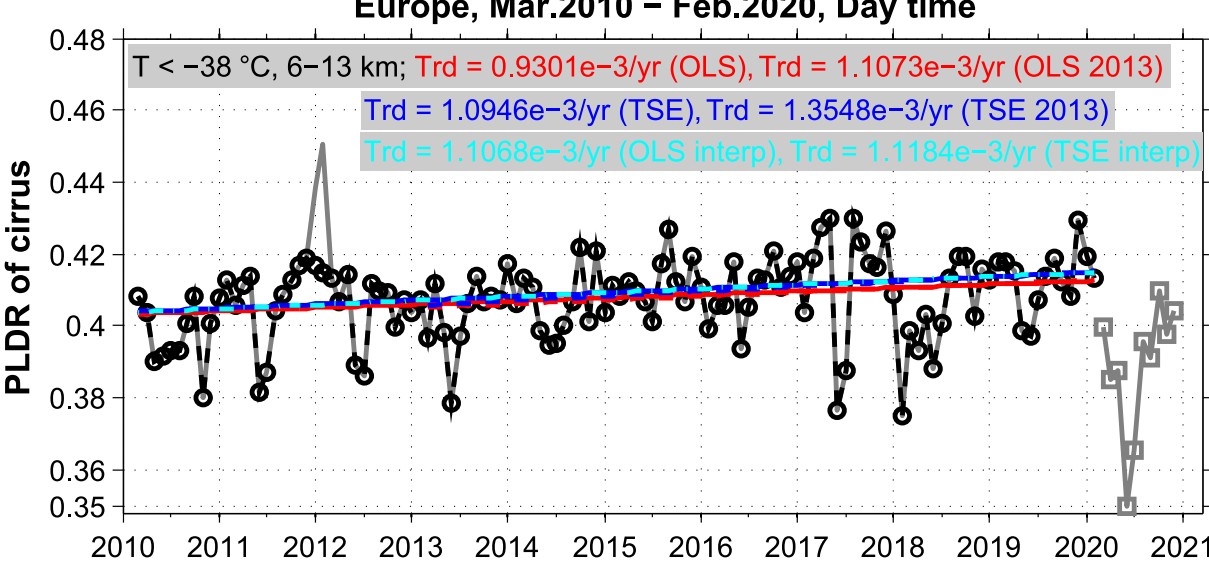

**Figure 5.** Medians of cirrus $\delta_p$ values in different month from March 2010 to February 2020 shown in gray lines and during the COVID-19 pandemic starting from March 2020 shown in squares. Both ordinary least square (OLS) estimator and Theil-Sen estimator (TSE) methods are applied to determine the long-term trend (i.e., slope) of the cirrus $\delta_p$ (medians) for the 10-year observations before COVID and the linear fits are shown with solid lines in red and blue, respectively. Furthermore, the same calculations are also carried out on the observations during the period from March 2013 to February 2020 (indicated also in the insert of the plot), when the aviation densities grew more strongly, and the corresponding fits with dashed lines in red and blue, respectively. Further, the outliers in January, February, and July 2012 are removed and the interpolated data sets are also conducted with the TSE method for comparison. The estimated long-term trends for different cases are indicated on the plot (in light blue). Upper panel shows the results from both day- and night-time observations and Lower panel from only day-time observations.





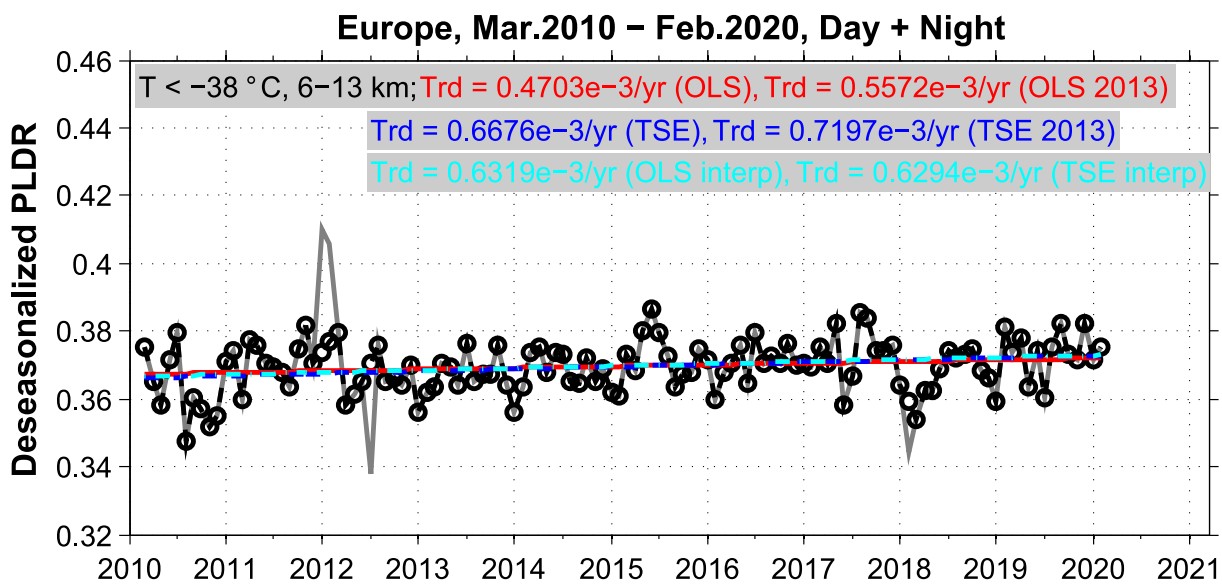

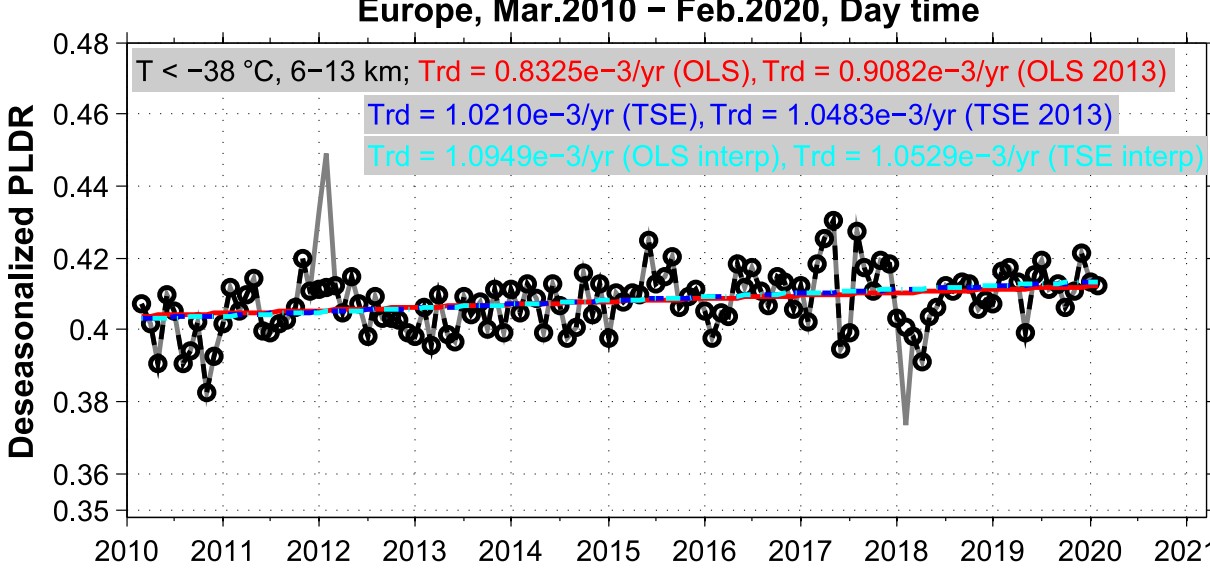

**Figure 6.** Same as Figure 5, but for the deseasonalized values of the cirrus PLDR.





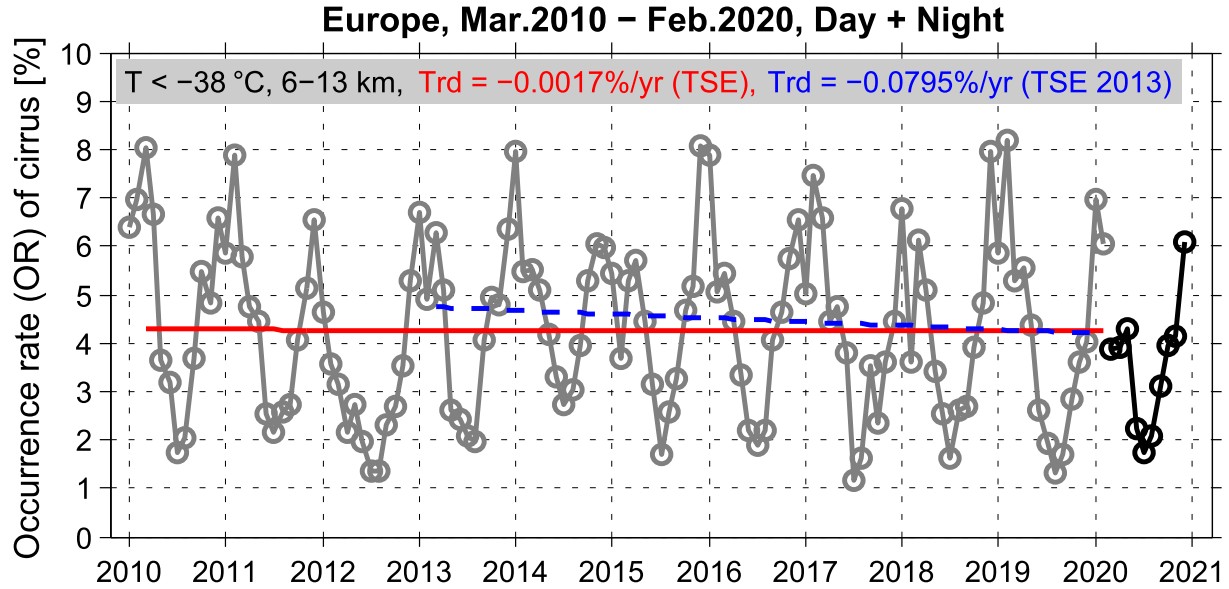

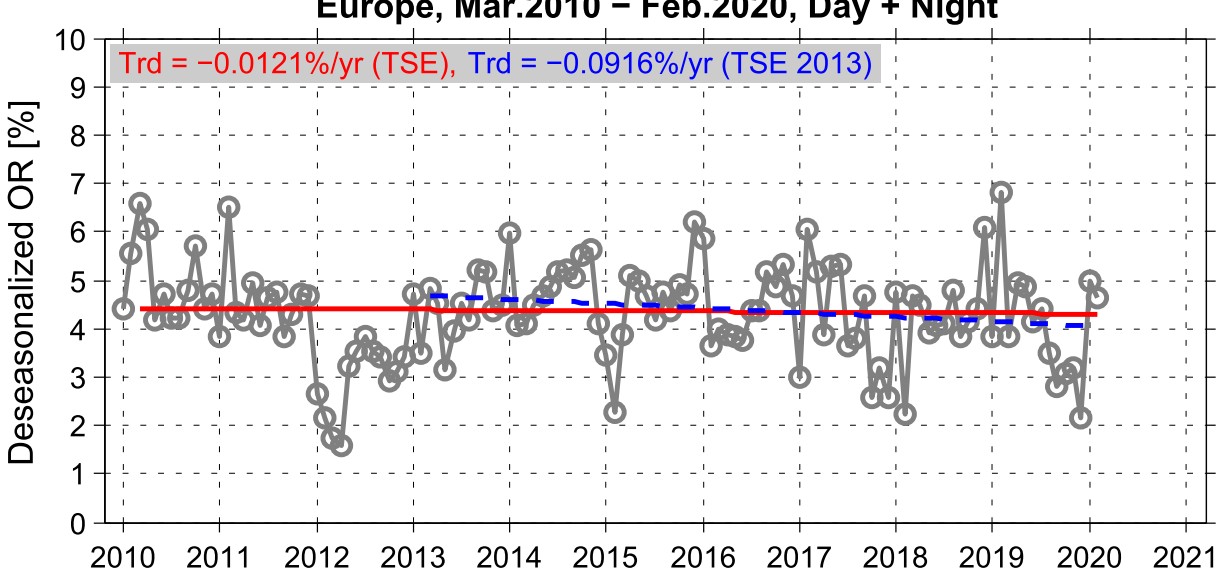

**Figure 7.** Long-term variation of monthly occurrence rate of cirrus clouds (OR) in years from 2010 to 2020 as well as the linear fitting for the values in the periods from Mar.2010 to Feb. 2020 (in red) and from Mar.2013 to Feb. 2020 (in blue). The corresponding trends (i.e., the slopes) are indicated on the plot: Upper panel for the derived monthly OR from all the data including day- and night time; lower panel for the deseasonalized time series of OR.





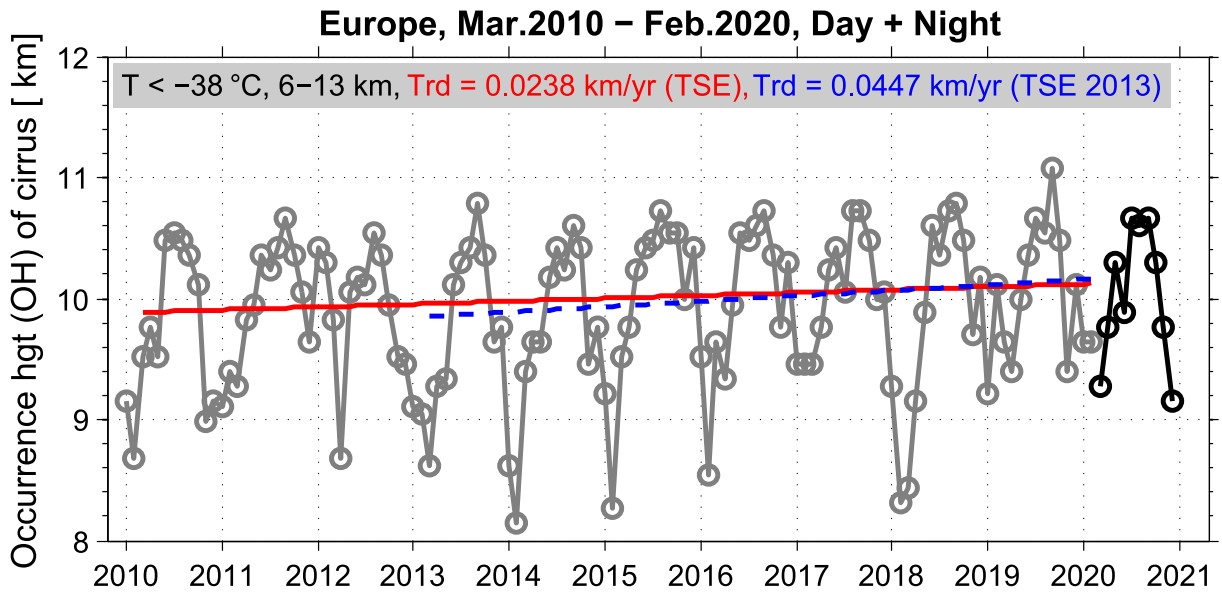

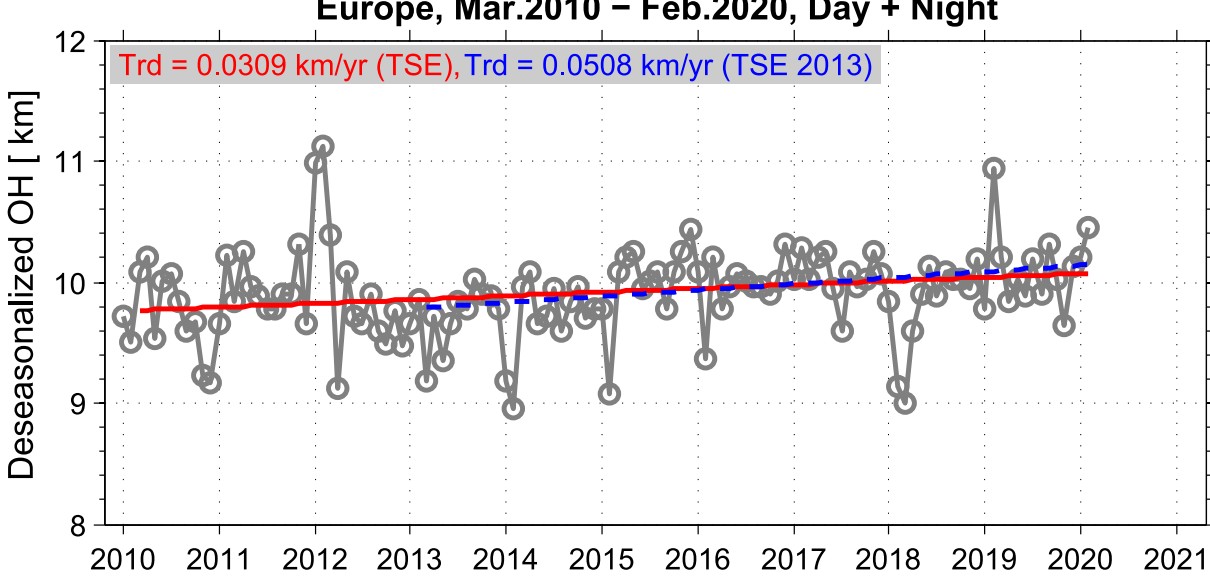

**Figure 8.** Long-trend variation of the occurrence height of cirrus clouds in the period from Mar. 2010 to Feb. 2020.







**Figure 9.** Correlation between the particle linear depolarization ratio (PLDR) $\delta_p$ of cirrus clouds and the ambient temperatures derived from the lidar meausurements of CALIPSO and GEOS-5 model data, respectively, over the European region during the period from March 2010 to February 2020. A linear fitting conducted for all the data (shown in black) shows a negative correlation between them. Further, a more significant correlation (negative) can be estimated between both quantities at the lower temperatures below -50°C but very small, if not no, correlation at higher temperatures above, which is the same case for the data of each individual month (Li and Groß, 2021). The color codes are used to visualize the number densities of scatter point data.





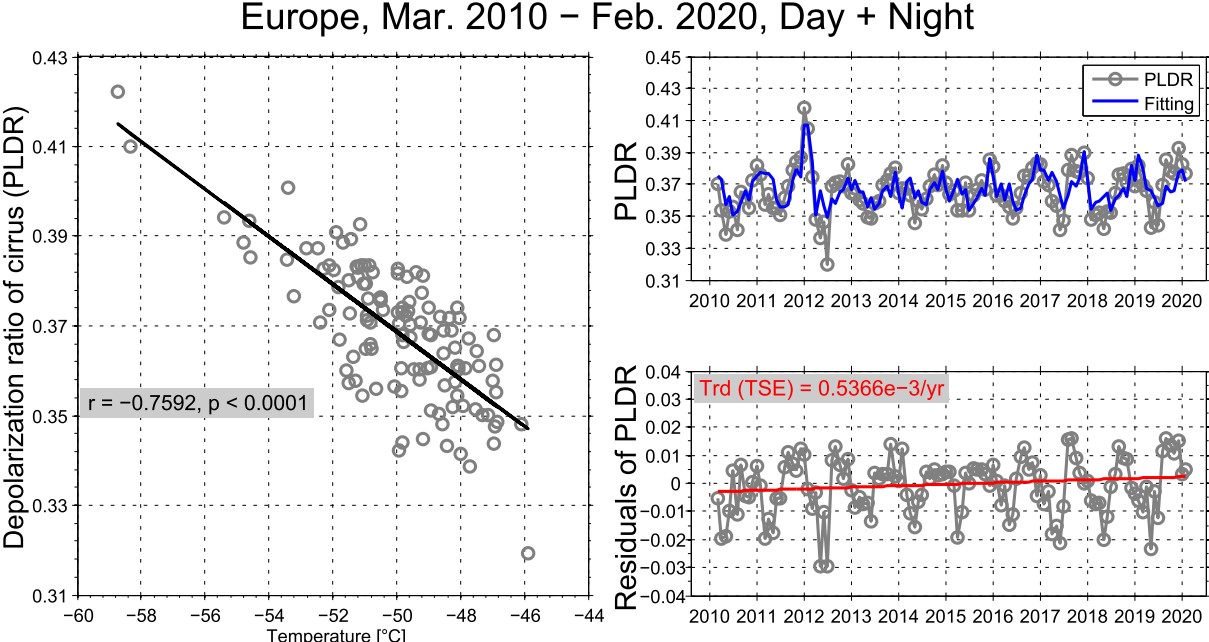

**Figure 10.** Left panel: The gray circles stand for the correlation between the monthly medians of cirrus $\delta_p$ and the corresponding ambient temperatures and the black line for the linear fitting line derived from all data points. Right panels: the monthly medians of cirrus $\delta_p$ in the period from Mar. 2010 to Feb. 2020 are shown in gray and the regressed values with the first degree polynomial in blue (Right-upper panel); The residuals after removing the regressed values from the monthly medians of $\delta_p$ are shown in gray and the linear fitting line with the Theil-Sen estimator (TSE) method in red, with the long-term trend of 0.54e-3/yr indicated on the plots (Right-lower panel). However, the significance test with the Mann-Kendall trend test shows that the trend in the residuals of $\delta_p$ removing the temperature contribution is at 90% confidence level (with $p = 0.0582$).



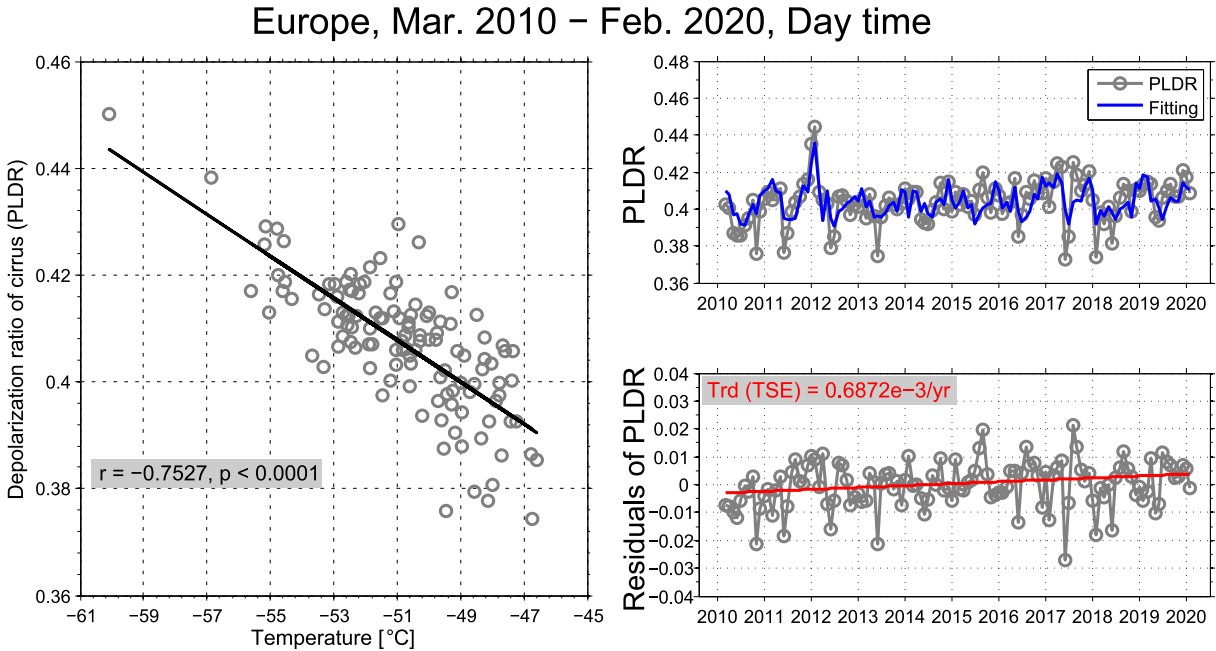

**Figure 11.** Same as Figure 10, but for the day-time observations. There is a trend of 0.69e-3/yr in the residuals of $\delta_p$ removing the temperature contribution which is statistically significant with $p < 0.001$ according to the Mann-Kendall trend test.



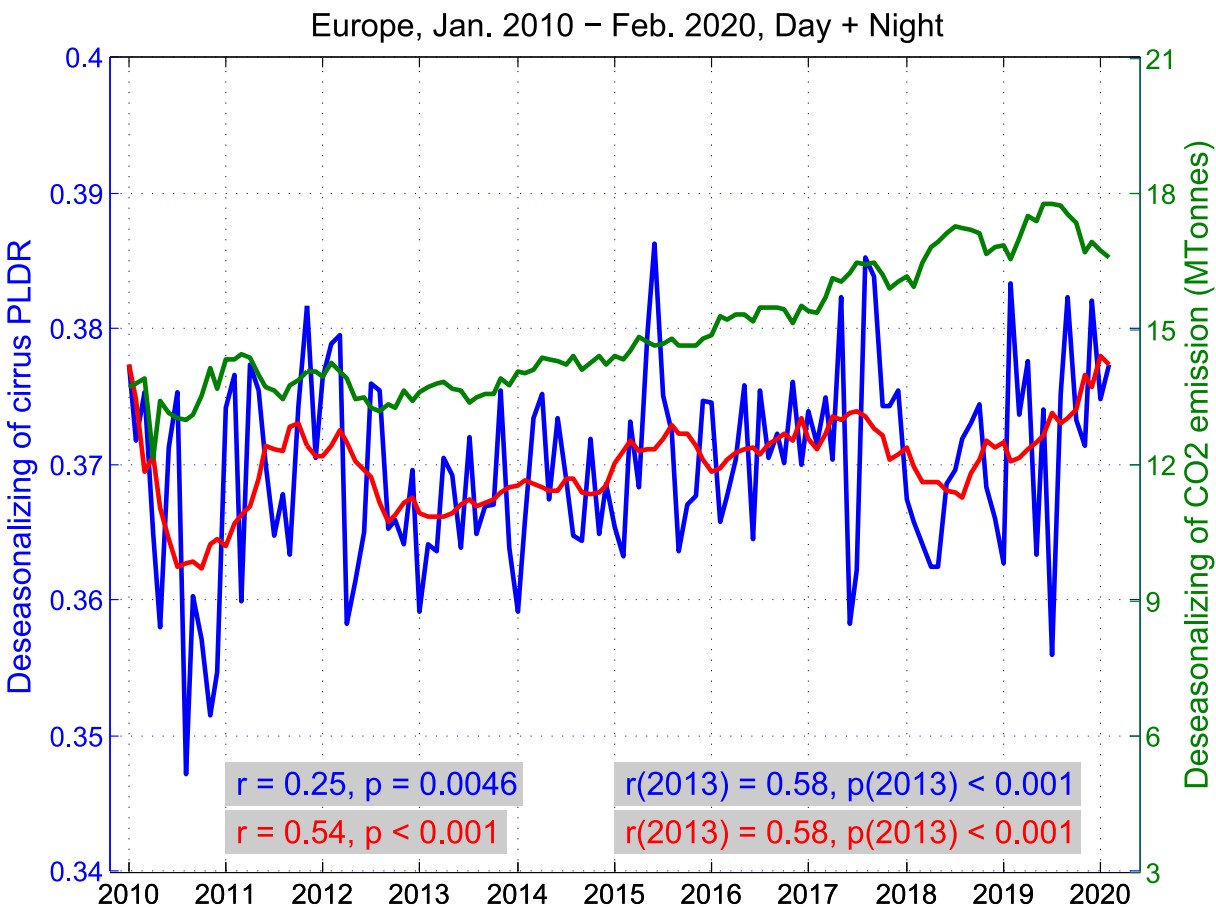

**Figure 12.** Comparison between the deseasonalized time series of $\delta_p$ (in blue) and the corresponding deseasonalized CO2 emissions from aviation in Europe (in green). The annual mean variations of the deseasonalized $\delta_p$ are overplotted with a 12-point moving average smooth. The correlation coefficients between both parameters for a full period as well as for a shorter period from January 2013 are calculated, which are all at the confidence level above 99.5%.





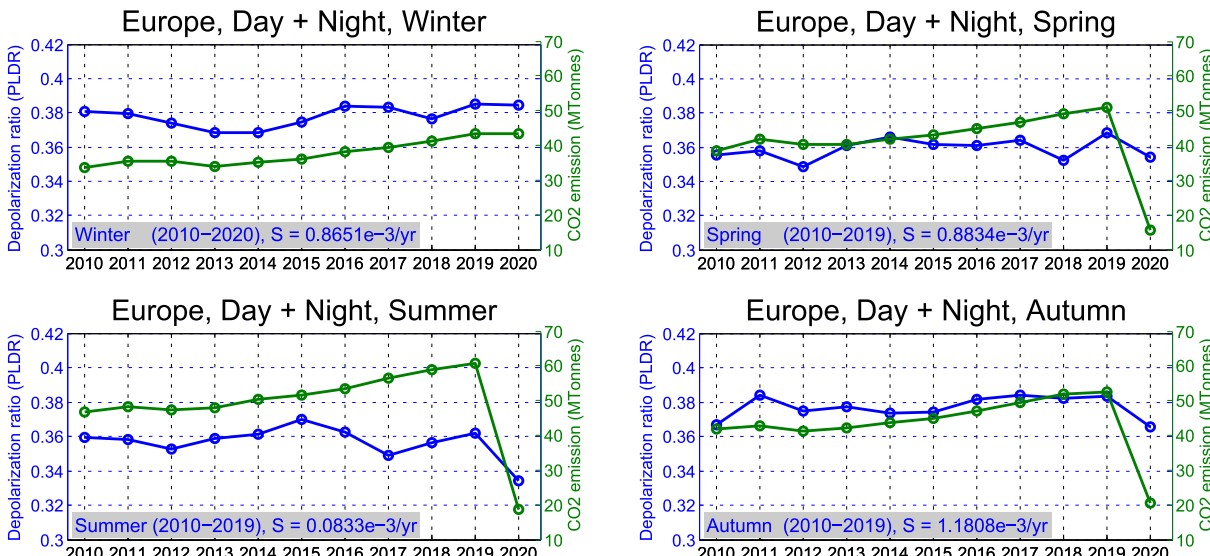

**Figure 13.** Long-term trends in cirrus $\delta_p$ in different seasons and their correlations with the corresponding CO2 emissions from aviation in Europe. There are increasing trends in $\delta_p$ as well as CO2 emissions from aviation in all the seasons in the 10 years before COVID. The correlation analysis between both parameters shows a strong correlation in winter, spring, and autumn, but a weak correlation in summer.