# Peer review of "Satellite observations of seasonality and long-term trend in cirrus cloud properties over Europe: Investigation of possible aviation impacts"

_EGUsphere, 2022_

## Referee Comment (RC2)

[referee-annotated manuscript omitted]

---

## Author Comment (AC1)

**Response to Review RC1 by Referee #1.**

We thank the Referee for his/her careful review and for the comments and suggestions for revisions. In the following, **the Referee's questions and comments are repeated in black and our responses follow in blue.**

In this work 10-year lidar measurements of cirrus clouds with CALIPSO are analyzed to determine inter and intra annual variability and trends of their occurrence and optical characteristics, namely depolarization. Correlation with atmospheric temperatures and air traffic are also explored.

A seasonal cycle was detected both in occurrence and optical properties, with larger values of both parameters in winter A positive trend in the deseasonalized depolarisation time serie and a negative trend in the deseasonalized occurrence rate time serie were also demonstrated.

As the author claims that there exists a positive trend in air temperature at cirrus altitude, and since depolarization is temperature dependent to a certain extent, they remove such effect from the depolarization time series. This is done by applying a linear regression model on the depolarization-temperature dependence and subtracting the model from the depolarization time series. The time series of the depolarization residuals again shows a positive trend. The author link this positive trend to an increase of air traffic over the time window of the dataset, this latter estimated from the increase in its contribution in CO2 emissions.

The work is interesting and important and certainly deserves to be published. However, there are two aspects that, in my opinion, need further study before publicaton.

➔ Thank you for the general comments.

First, I must note that the work it is not fully convincing in studying the interannual trend of air temperatures. In fact, while the seasonal trend of temperatures is clear, a decadal trend is not. I think this is the biggest problem of the work, which otherwise demonstrates a sufficient maturity to ensure its publication. For this reason, I encourage the authors to dwell more on the study of the decadal trend of the air temperature, to accompany their findings with any supporting literature or otherwise to make their claims on such topic less assertive.

➔ The reviewer is right that the seasonal trend of temperatures is clear but a decadal one is not. Sorry for the misleading. The manuscript does not focus on the trends in air temperatures or humidity (either seasonal or decadal trend), but on the trends in the cirrus properties (OR and PLDR). They for sure depend on the meteorological conditions (incl. temperatures). The decadal evolutions of temperature, humidity, and wind fields shown in Figure 2 are used to present a general picture of the meteorological conditions in the research area. The derived trends in the meteorological data (unlikely statistically significant) just provide us information of the background which are in general stable for the period we focus on. The corresponding revisions have been made on the manuscript.

A second aspect that has not been sufficiently analyzed concerns the lack of an analysis if the observed trends are due to only to the increase in cirrus originated by contrail, or rather to changes in the microphysics of cirrus clouds in general. I do not know (I doubt with sole lidar data) if there is the possibility of dividing the dataset into two categories, based on the proximity of the observations to air corridors, or on the vertical and/or optical thickness of the cirrus

clouds, or their horizontal (i.e. along satellite track) extent in order to identify the cirrus clouds originating from contrail with respect to the "natural" others. If such categorization were possible, the work would undoubtedly benefit from a non-aggregated analysis. In any case, I believe that the question should still be discussed.

➔ The reviewer is right that it is difficult (likely impossible) to distinguish the cirrus clouds originated by contrail from natural cirrus clouds using the CALIPSO lidar measurements only. The origins of cirrus clouds can't be identified according to their microphysical properties, because contrails at different phases consist of ice crystals with different size and shapes (initiating from young line-shaped contrails transforming into cirrus-like contrails; e.g., Burkhardt and Kärcher, 2011; Iwabuchi et al., 2012). The young contrails (< 2 min) containing quasi-spherical ice crystals are characterized by low PLDR and the persistent irregularly-shaped contrails or contrail cirrus by high PLDR. The point of the changes due to the lack of contrails is a bit uneasy. Therefore, independent information, like flight tracks of aircrafts, is needed to identify contrails from natural cirrus clouds (e.g., Tesche et al., 2016). In our previous study (Li and Groß, 2021), we could show that the changes in the cirrus studying PLDR are found throughout the whole altitude range of cirrus clouds. Further, we have shown in another paper (Groß et al., 2022) that young contrails (at least when embedded in natural cirrus clouds) do not have those large differences compared to the changes of the optical properties due to the aviation impact.

One final note, I as a non-native English speaker, found English of the text rather tiring to read. I would suggest having the text proofread by a native English speaker.

➔ Sorry for the bother. We will try to polish the languages of the manuscript.

In the following, moredetailed annotations on the text (page, line)

(1,4) "…are supposed…"

➔ Changed, thank you.

(2,56) typo "depdendence"

➔ Changed, thank you.

(3,68-69) Unclear. Could rephrase as : "The light emitted by lidar AND BACKSCATTERED BY SPHERICAL PARTICLES exhibit the same orientation of polarization as the incident light if it is scattered WHILE THE POLARIZATION CHANGES and different polarization if scattered WHEN THE LIGHT IS BACKSCATTERED by non-spherical particles such as cirrus ice crystals."

➔ The suggestions by the reviewer are not clear in our opinion and won't be accepted. However, the sentence has been rephrased anyway.

(3,71) I would cancel , "e.g., non-spherical mineral dust particles with high values of δp" or add other types of non-spherical aerosol that have different mean values of δp (Biomas Burning aerosol, Sea Salt, etc..)

➔ Canceled as suggested, thank you.

(3,75) Pristine habits are mainly driven by the temperature at which a crystal forms and, maybe to a lesser extent, by the humidity of the air. However the internal cloud dynamics and lifetime duration and stage strongly influence the shape of crystals as well. That should be mentioned.

➡ Added as suggested, thank you.

(4,112) Typo „difficulat"

➡ Corrected, thank you.

(4,112-113) „However, there is an aviation fingerprint with two maxima during eastbound and afternoon westbound traffic in the area we are focusing on here." maximum of what? what is the area we are focusing on?

➡ We meant the maxima of aviation fingerprint (or flight densities). The area we are focusing on here is the north Atlantic air traffic corridor that covers the most part of the research area of this current manuscript. We have made revisions correspondingly.

(4,114) „Therefore…" the lack of clarity of the previous sentence makes its consequences unclear.

➡ The previous sentence clarifies that there are two maxima of flight densities during the morning eastbound and afternoon westbound traffic, which implies that aviation effects on cirrus clouds are expected to be stronger on day-time than night-tine. We have made revisions accordingly. Thank you.

(4,116-118) This should be shifted upward.

➡ The sentence has been shifted upward and the rearrangements have been done, which, however, is not a better choice. Anyway, thank you.

(5,135) „extreme lower". Lowest?

➡ No, "extreme low temperatures" is the right expression that I want to say here. But thank you.

(5,139-140) Could you comment on the statistical significance of such results? At face value it does not seem high.

➡ We did compare the year-to-year variabilities of air temperatures in different seasons (see the plot below). The results clearly show that temperatures increased more significantly in winter than in summer.

[Figure]

(5,141-144) Please discuss the statistical significance of the findings.

➔ A significance test has been done for the derived trend in the RHi which, however, is not statistically significant. The corresponding test results have been added in the manuscript.

(5,149-153) I honestly don't think this assertion is sufficiently supported by the above analysis. Tests for the presence of trend, confidence intervals for the trend, etc. should have been performed. In the absence of such tests (which however I encourage the authors to conduct) I suggest reformulating the sentence in a less assertive form and / or recalling any studies in the literature supporting these same conclusions.

➔ The reviewer is right. The significance tests should be done for the derived trends. It's done and the corresponding confidence levels for each have been added in the description.

(5,156) To my knowledge, the common cruising altitude for most commercial airplanes is between 10 and 13 km . typically, aircraft fly around 10-11 km.

➔ Thank you for pointing out. For the North Atlantic air traffic corridor connecting North America and Europe, the cruising altitudes are between 8.8 and 12.5 km. Please see https://en.wikipedia.org/wiki/North_Atlantic_Tracks
Do you Maybe have any other sources on the information of typical cruising altitudes of aircrafts? Thank you.

(6,175) "Please note…" How do you exclude deep convecton cirrus? Please specify the methodology.

➔ The CALIPSO science team developed the Vertical Feature Mask (VFM) giving information on the nature of targets in the lidar probing profiles with a quality flag.  The cloud layers are distinguished from aerosols and further according to the standard meteorological cloud types defined by the ISCCP Cirrus clouds are further distinguished from water-phase clouds and ice clouds due to deep convections (see references Winker et al., 2009; Liu et al., 2009).

(10,288-295) "Generally…temperatures." As in the subsequent analysis the nonlinear regression models have not been used, there is no need to quote them here. Unless you provide a justification for the choice of the linear instead of the non linear one.

➔ The reviewer is right. We have removed the equation of nonlinear regression and rephrased accordingly. Thank you.

(13,354) See my comment on (5,149-153).

➔ Thank you for pointing out. The sentence is not correctly placed and removed.

**Reference**

Groß, S., Jurkat-Witschas, T., Li, Q., Wirth, M., Urbanek, B., Krämer, M., and Voigt, C.: Investigating an indirect aviation effect on mid-latitude cirrus clouds – linking lidar derived optical properties to in-situ measurements, Atmos. Chem. Phys., in review, 2022.

Iwabuchi, H., Yang, P., and Liou, K. N., and Minnis, P.: Physical and optical properties of persistent contrails: Climatology and interpretation, J. Geophys. Res., 117, D06215, https://doi.org/10.1029/2011JD017020, 2012.

Li, Q., and Groß, S.: Changes in cirrus cloud properties and occurrence over Europe during the COVID-19-caused air traffic reduction, Atmos. Chem. Phys., 21, 14573–14590, https://doi.org/10.5194/acp-21-14573-2021, 2021.

Tesche, M., Achtert, P., Glantz, P., and Noone, K. J.: Aviaiton effects on already-existing cirrus clouds, Nat. Commum., 7, 12016, https://doi.org/10.1038/ncomms12016, 2016.

---

## Author Comment (AC2)

**Response to Review RC2 by Referee #2.**

We thank the Referee for carefully reading our manuscript and for the comments and suggestions to improve the current work. In the following **the Referee's comments are repeated in black and our responses follow in blue.**

The paper Satellite observations of seasonality and long-term trend in cirrus cloud properties over Europe: Investigation of possible aviation impacts by Qiang Li and Silke Groß is highly interesting and robust.

It is relevant in showing how vertical profiles from satellites could give a better insight of the atmospheric estate, fill gaps in knowledge, and pose new scientific questions.

As general comments, I think it is detailed and many investigations are reported. The impression is that the reader can sometimes be lost in the progress of such reporting. I would suggest reducing the number of figures and focusing more on what is the main result. Figure 12 is the main message that probably authors would like to give as a take-home message, but this is somehow diluted by the presence of many analyses: these are relevant for reaching the main results but could be shortened and eventually reported as an appendix or additional material.

- ➜ Thank you for the general comments.
- ➜ It is stated in the manuscript that the cirrus morphologies and occurrence rates as well as the high degree of variability in their microphysical properties highly depend on the substantial differences in the meteorological conditions. To draw a final conclusion of the aviation impact on cirrus properties, it is necessary to discuss as detailed as possible how the cirrus properties vary depending on the meteorological conditions and their occurring heights. According to the comments, we have tried to reduce the figures of the results from the day-time observations and kept the description of them only in text.

Apart from this general comment, 3 are the points to be clarified /discussed/fixed in the paper:

- It seems that 2 different models are used for temperature and humidity during the investigation: ECMWF and GEOS. Why this difference? Why not use the same for the 2 analyses reported? Please clarify

=> The CALIPSO science team determine meteorological conditions (including temperatures) from the GEOS-5 data assimilation products (from GMAO) interpolated in space and time to the CALIPSO orbital tracks. The data are saved along with other original measurements of targets (including aerosols and clouds) by the onboard instruments. It is robust to use the coordinated data of meteorological conditions for comparisons with the cirrus properties. While in Figure 2 of the manuscript, the ERA5 data are used to give a general picture of the meteorological conditions of the research area including temperature, humidity, and wind fields. ECMWF is considered as the most advanced and most reliable model. In comparison to the GEOS-5 data, the ERA-5 data have a better resolution and accuracy of the forecast. Furthermore, they are run based on different data assimilation process and governing equations, the archive of meteorological information from one model in different year should be consistent. A general picture of meteorological conditions is enough for our analysis and should not bring any misleading.

- In the PLRD temporal behavior of fig 12, there is an anomaly in the 2010 and 2017-2019 (mainly 2018) period: is it possible that the big volcanic eruption affecting Europe in 2010 is

the cause of the 2010 anomaly? Is the aerosol/cloud misclassification in VFM a potential issue then? Which could be the reason for the lower PDLR in 2017-2018? Please discuss this point

=> Many flights in Europe were canceled because of the volcanic eruption in 2010. But the air travel disruption only lasted for a short term. From Figure 1 of the manuscript, the air traffic in Europe (incl. 42 countries and regions) in 2010 from April on did not show big anomaly (departure) from the corresponding periods in other years. The deseasonalization process was done on the data by computing the monthly climatological mean, subtracting them from each monthly record and finally adding the total mean of $\delta p$. The extreme values of $\delta p$ might change the deseasonalized dataset significantly. The unexpected low values of deseasonalized $\delta p$ in the last second half year of 2010 and first half year of 2018 are supposed to be due to the lower values of $\delta p$ in August and November 2010 as well as in February-March 2018 compared to the corresponding months in other years, respectively. This has been discussed in the text that the occurring heights of cirrus are supposed to be correlated with the extreme values (see Figure 7). According to the height dependence of PLDR, the departures of PLDR in the mentioned months will lead to the anomalies in the deseasonalized datasets. The corresponding discussions have been added in the manuscript. Thank you for the comments.

- I am not a native English speaker, but the paper is somehow hard to read. I reported some revisions in the comments in the attached pdf, but these are just examples. Please revise the paper carefully in this sense.

=> Sorry for the bother. We will try our best to polish the languages of the manuscripts.

These and more detailed points are reported as comments in the pdf file.

=> The pdf file is the original manuscript.

---

## Author Response (AR2)

Dear Matthias,

Thank you very much for the comments. Followings those, we have made the corresponding revisions. Figure 8 of the previous version was removed and the main information of the plot has been described in text. The plot titles were removed. The captions have been accordingly revised with the descriptions moved to the corresponding sections. For the symbols, we kept PLDR instead of $\delta p$ to make a consistence with our previous work.

Thanks again for the editing of this manuscript.

Best regards

Qiang